

# Streamflow generation in a nested system of intermittent and perennial tropical streams under changing land use

Giovanny M. Mosquera[1], Daniela Rosero-López[1], José Daza[1], Daniel Escobar-Camacho[1], Annika Künne[2], Patricio Crespo[3], Sven Kralisch[2], Jordan Karubian[4,5], Andrea Encalada[1]

[1]Laboratorio de Ecología Acuática, Instituto BIOSFERA, Universidad San Francisco de Quito USFQ, Quito, Ecuador
[2]Geographic Information Science Group, Institute of Geography, Friedrich Schiller University Jena, Jena, Germany
[3]Departamento de Recursos Hídricos y Ciencias Ambientales & Facultad de Ingeniería, Universidad de Cuenca, Cuenca, Ecuador
[4]Department of Ecology and Evolutionary Biology, Tulane University, New Orleans, LA, USA
[5]Fundación para la Conservación de Los Andes Tropicales, Quito, Ecuador

*Correspondence to*: Giovanny M. Mosquera (giovamosquera@gmail.com); Andrea C. Encalada (aencalada@usfq.edu.ec)

**Abstract.** Despite the increased interest in the hydrology of intermittent hydrological systems in recent years, little attention has been given to tropical forest environments. We present a unique set of hydrological, stable isotopic, geochemical, and landscape mapping information to obtain a mechanistic understanding of streamflow generation in an intermittent system of 20 nested catchments (<1-159 km$^2$) draining intermittent and perennial streams and rivers in the Chocó-Darien ecoregion, a tropical biodiversity hotspot, located in the Pacific lowlands of northern Ecuador that has been strongly degraded by deforestation and cultivation during the last half-century. Intermittent streams mainly located in conserved forested headwaters present a faster streamflow response to rainfall and shorter recession times than degraded perennial streams in the catchment's middle and lower parts. Isotopic information shows that rainfall during the wet period (January to May) contributes to streamflow generation in intermittent streams possessing shallow soils and a low bedrock permeability, in contrast to perennial streams in which rainfall during the wet season recharges their high bedrock permeability. Lower concentrations of major ions and electrical conductivity were observed in intermittent streams compared to higher concentrations in perennial streams. We found a strong correlation between the catchments' geology and their geochemical signals and a weak correlation with their topography, land cover, and soil type. These findings indicate that shallow subsurface flow paths through the organic horizon of the soil dominate streamflow generation in intermittent streams due to the limited water storage capacity of their bedrock with very low permeability. On the contrary, high bedrock permeability increases the water storage capacity of perennial catchments replenished during the wet period, helping sustain streamflow generation throughout the year. These findings highlight the key role geology plays in driving hydrological intermittency, even in highly degraded tropical environments, and provide key process-based information useful for water management and hydrological modelling of intermittent hydrological systems.



## 1 Introduction

Intermittent hydrological systems are defined as drainage areas that partially or totally cease to flow temporarily during part of the year (Meerveld et al., 2020; Shanafield et al., 2021). Those systems account for more than half the length of the global drainage network and this proportion is expected to augment due to changes in land use and global climate (Messager et al.,

2021). The streamflow dynamic of intermittent hydrological systems generally varies markedly between high flows during wet periods and no flow during dry ones, resulting in important socioeconomic, ecological, and biological implications (Costigan et al., 2017). As a result, flow from intermittent hydrological systems provides key ecosystem services, including water provision to humans, environmental water quality, and the preservation of life in aquatic ecosystems (Pastor et al., 2022). As such hydrological dynamics can be caused by one or the combination of several factors, including

hydrometeorological conditions, surface and subsurface catchment characteristics (i.e. topography, soils, geology), and land cover (e.g. groundwater extraction, deforestation; Costigan et al., 2016); obtaining a mechanistic (process-based) understanding of streamflow generation in intermittent hydrological systems is paramount to improving their conservation and sustainable management (Leigh et al., 2016; Vander Vorste et al., 2019) and to develop accurate predictive models of hydrological intermittency to assess the impacts of global change drivers (Dohman et al., 2021; Shanafield et al., 2021). This

knowledge is particularly needed in tropical regions that host the largest proportion of the planet's biodiversity (Myers et al., 2000) and are drastically affected by changes in land use due to fast population growth (Newbold et al., 2020).

Despite the social-ecological importance of intermittent hydrological systems generally composed of a combination of intermittent and perennial streams (Meerveld et al., 2020), field-based studies that allow obtaining a sound mechanistic understanding of streamflow generation in these systems remain limited worldwide (Shanafield et al., 2021). Indeed, only a

few studies have investigated the sources and flow paths of water influencing streamflow generation in intermittent rivers and streams. For instance, streamflow generation in an intermittent low relief and highly weathered catchment with a subtropical climate in North Carolina, USA, was investigated by Zimmer and McGlynn (2017). The authors found that two different water flow paths influencing hydrograph recession were activated depending on seasonal evapotranspiration. In semi-arid southeast Australia, Zhou and Cartwright (2021) quantified the sources of baseflow in an intermittent catchment.

They determined that near-river water storage was the main source of baseflow as opposed to regional groundwater. In South Africa, Banda et al. (2023) assessed the interaction of surface water and groundwater in an intermittent catchment with Mediterranean climate. They found that geology plays a key role in surface-groundwater hydrological connectivity. Bourke et al. (2021) also reported that bedrock permeability controls streamflow generation in a low-relief and semi-arid region in northwestern Australia. The flow paths influencing surface flow in an intermittent stream section in a semi-arid climate in

the Rocky Mountains, Idaho, USA was evaluated by Dohman et al. (2021). The authors concluded that flow cessation preferentially occurred where hillslopes did not laterally contribute to streamflow and where greater vertical losses from the stream bed to groundwater were observed. Streamflow generation in a tropical dry forest catchment near the Pacific coast of central Mexico was investigated by Farrick and Branfireun (2015). They reported that vertical water flow paths and the





contribution of water stored in the saturated zone of the catchment were the most influential factors driving streamflow
generation. Although these investigations have helped to better comprehend streamflow generation in intermittent
hydrological systems, this understanding remains elusive in highly seasonal tropical forest environments. This lack of
understanding not only limits water management in intermittent hydrological systems, but also the assessment of how their
hydrological dynamic compares to that in non-tropical environments.

One of the key factors limiting the advancement in the understanding of streamflow generation in intermittent hydrological
systems is the difficulty of collecting data of different nature to produce robust conceptualization of flow processes (e.g.
Jacobs et al., 2018; Mosquera et al., 2016a; Timbe et al., 2017). While hydrometric (e.g. water level or discharge) data alone
allow characterizing the dynamic of streamflow and its characteristics (Willems, 2014; Zimmer and McGlynn, 2017), such
data does not provide information about the sources and main flow paths of water contributing to streamflow generation
(Leibundgut et al., 2009; McDonnell and Kendall, 1992; Mosquera et al., 2016a). Isotopic (e.g. water stable isotopes) and
geochemical (e.g. dissolved elements) tracers as well as water physicochemical parameters (e.g. water temperature and
electrical conductivity) have been used to identify water sources (e.g. Barthold et al., 2010; Correa et al., 2017; Inamdar et
al., 2013; Liu et al., 2004; Penna et al., 2014) and define the main flow paths water follows through catchments as rainfall
becomes streamflow (e.g. Asano et al., 2002; Lahuatte et al., 2022; Laudon et al., 2004; Mosquera et al., 2020; Muñoz-
Villers and McDonnell, 2012; Ramón et al., 2021; Tetzlaff et al., 2015) in humid environments. In addition, mapping of
biophysical landscape characteristics using nested monitoring systems (as opposed to single or paired catchment approaches)
has proven useful to understanding the influence of catchment intrinsic features such as topography, soils, geology, and land
cover in hydrological systems where streamflow is perennial (e.g. McGuire et al., 2005; Mosquera et al., 2015, 2016b;
Muñoz-Villers et al., 2016; Staudinger et al., 2017). The clear benefits of employing multimethod approaches that combine
hydrometric, isotopic, geochemical, and landscape mapping, among other data sources in humid, perennial environments,
suggest their great potential to achieve a sound process-based comprehension of intermittent hydrological systems (e.g.
Banda et al., 2023; Bourke et al., 2021; Farrick and Branfireun, 2015). Thus, the use of such approaches to investigate
streamflow generation in those systems should be encouraged, particularly in regions where hydrological information is
scarce but urgently needed for timely management of water resources and aquatic ecosystems, including understudied
tropical areas.

Considering the lack of studies on streamflow generation in intermittent hydrological systems in tropical regions, particularly
in South America (Shanafield et al., 2021), we aim to fill this knowledge gap. To this end, we apply a multimethod approach
to a unique data set including hydrometric, isotopic, geochemical, and landscape mapping information collected through a
nested monitoring system comprising 20 intermittent and perennial streams within the Cube River catchment located in the
Ecuadorian Pacific lowlands of the Chocó-Darien ecoregion. Since the studied catchment has undergone strong deforestation
and agricultural activities in the last 50 years, it is representative of regional land use change (MAATE, 2022). Therefore, the
monitoring setup in this land use template allows us to pose a regional overarching question: what are the main water flow
paths influencing streamflow generation in a nested system of intermittent and perennial tropical streams under changing



land use? On the one hand, addressing this question will add to the overall mechanistic understanding of the hydrology of intermittent rivers and streams to develop robust models of hydrological intermittent regimes at different spatial and
temporal scales. On the other hand, this knowledge will also provide key information to assess water resources availability, manage and conserve aquatic ecosystems, and evaluate the resilience of the system to global change stressors.

## 2 Study area

The study was conducted at the Cube River catchment (Fig. 1). The catchment is a tributary of the Esmeraldas River Basin that subsequently drains into the Pacific Ocean. It situates in the Pacific lowlands of northern Ecuador (0.37°N, 79.66°W)
within the Chocó-Darien ecoregion (Dinerstein et al., 2017), a worldwide biodiversity hotspot extending from southern Panama to northern Ecuador (Myers et al., 2000). The geological and climatological configuration of the ecoregion provides a myriad of ecosystems distributed between coastal ridges and the Andean cordillera that modulates regional and local climate patterns favouring exuberant vegetative formations, endemic species, and vast ecosystem services (Fagua and Ramsey, 2019). Part of the Cube River catchment situates within this protected area that is surrounded by two mountain
ranges: the Mache and the Chindul ridges (200-800 m. a.s.l.), a coastal massif approximately 100 km west of the Andes (hereafter referred to as the Mache-Chindul Mountain range). The catchment drains from south to north, encompassing a drainage area of 159 km$^2$, with a mean slope of 32%, and a mean altitude of 339 m a.s.l. ranging from 53 and 688 m a.s.l. (Fig. 1a). The study area presents a strong hydroclimatological variability with a wet period extending from January to May and a dry period from June to December (Molinero et al., 2019). This seasonal hydrometeorological condition results in the
occurrence of hydrological intermittence, particularly in headwater tributaries. Land cover in the catchment is representative of the region in which exuberant tropical forests have been degraded by human activities during the last half-century (Cuesta et al., 2017). Due to deforestation, cultivation, and cattle grazing, the Chocó-Darien ecoregion is particularly threatened in Ecuador. As a result, an agricultural mosaic comprising cattle grazing and cultivation land covers 69% of the catchment, with only 28% of native (e.g. primary and secondary) forest land remaining (Fig. 1b). The upper part of the catchment is
situated within private protected reserves for land-forest conservation, including Bilsa, Fundación para la Conservación de los Andes Tropicales (FCAT), and the national protected area of the Mache-Chindul Ecological Reserve (Fig. 1b).

Soils in the study area are generally shallow regardless of the type: inceptisols, mollisols, entisols, and miscellaneous (Figure 1c; MAG, 2019). Inceptisols distributed along the whole catchment are the dominant soils covering 79% of the study area (Fig. 1c). These soils possess a slightly acid-to-neutral pH and have a clay to clay loam texture. Mollisols cover 11% of the
catchment and are occasionally found in the south and central parts of the catchment. These soils are slightly acidic and clayey. Entisols cover 7% of the study site and are sporadically found in the central part of the catchment. They present a moderately acid to neutral pH and a clay to clay loam texture. Undescribed soils classified as "Miscellaneous" cover less than three percent of the catchment area and are found as small patches in the central part of the catchment.



**Figure 1: Cube River catchment (159 km²) biophysical maps of a) altitude, b) land cover, c) soil type, and d) geology. e) Location of the Cube River catchment (red area) in the lower part of the Esmeraldas River basin (purple line) in northwestern Ecuador. Maps a) to d) show the Cube River catchment drainage network and the intermittent (yellow circles) and perennial (blue circles) sampling sites at the outlet of 19 nested subcatchments (S1 to S19) and the outlet of the catchment (S20) where hydrologic, geochemical, and isotopic data were collected during six monitoring campaigns during the period January-December 2021. The * symbol next to the name of the sampling sites in a) denotes sites where water level data were continuously monitored during the study period.**





The underlying bedrock belongs to two geological formations. The Playa Rica formation covers 11% of the catchment and is primarily found in headwaters areas (Fig. 1d). This formation dates from the Oligocene period (23-33 Ma). It possesses a
very low bedrock permeability (DIIEA, 2010) and its lithology is composed of shales and sandstones. The younger Viche formation dating from the Miocene (5-23 Ma) has a medium bedrock permeability. It covers 89% of the study area and is distributed across the remaining of the catchment. The Viche formation lithology mainly comprises silty clay with calcareous lenses, shales, and sandstones (DIIEA, 2010).

## 3 Data and methods

**3.1 Monitoring scheme and catchment features**

We used a nested monitoring system comprised of 20 sampling sites presenting a mixture of intermittent and perennial hydrological conditions within the Cube River catchment to collect hydrological, isotopic, and geochemical data from January to December 2021. Since no hydrological studies were previously carried out in the study area, the selection of sampling sites was carried out in collaboration with staff members of the FCAT reserve. The reserve staff mainly included
local community members who have ample knowledge of the area and have been actively participating in ecological and biological projects in the area for over 15 years. Such knowledge was invaluable in identifying appropriate sampling sites based on accessibility, security, and hydrological (intermittent versus perennial) conditions. The sites included 19 subcatchments (S1 to S19 in Fig. 1a) and the outlet of the catchment (S20). Eleven sites were classified as perennial and 9 as intermittent according to their drying patterns (Fig. 1).

As part of the study, we developed thematic maps of the catchment biophysical features (Figs. 1a-1d). The altitude map was constructed using the Alos Palsar Digital Elevation Model (30 m resolution) from the Japanese Aerospace Exploration Agency (Alos Palsar DEM 30m, Version 2.0, 2019). Due to the constant presence of clouds or fog across the catchment, we did not find a single satellite image showing the land cover of the whole study area. Thus, the land cover map was built using SENTINEL and PLANET satellite imagery (USGS, 2023) available during the period 2019 to 2022. It is worth noting that
due to the resolution of the satellite images and the similarity of the cultivation and cattle grazing signals, we could not readily distinguish between these land cover types, so they were aggregated and classified into a single category hereafter referred to as agricultural mosaic. Similarly, we could not distinguish between native primary and secondary forests, so both types of forests were aggregated into a single category of native forests. Soil and geological maps were also built using information from the Ecuadorian Ministry of Agriculture and Livestock (MAG, 2019). As shown in Figs. 1a-1d, the
sampling sites were spatially distributed across the catchment from the headwaters, which tend to present intermittent hydrological conditions to the middle and lower parts where flow is perennial. The main topographic, soil, geologic, and land cover characteristics of the 20 sampling sites are summarized in Table 1. This information was used to determine whether such features influence the catchment's hydrological behaviour.





**Table 1: Main landscape features of the 20 sampling sites monitored within the Cube River catchment.**

| Sampling site | Drainage area (km²) | Mean altitude (m a.s.l.) | Mean slope (%) | Land cover[a] (%) | | | | | Distribution of soil types[b] (%) | | | | Geology[c] (%) | |
|---|---|---|---|---|---|---|---|---|---|---|---|---|---|---|
| | | | | NF | MF | AM | WB | BG | MOL | ENT | INC | MIS | PR | VI |
| S1 | 0.1 | 534 | 15 | 60 | 34 | 4 | 0 | 2 | 0 | 0 | 100 | 0 | 100 | 0 |
| S2 | 4.3 | 525 | 30 | 65 | 0 | 35 | 0 | 0 | 65 | 0 | 33 | 2 | 32 | 68 |
| S3 | 0.1 | 504 | 14 | 94 | 0 | 0 | 0 | 6 | 0 | 0 | 100 | 0 | 100 | 0 |
| S4 | 0.1 | 600 | 21 | 13 | 0 | 87 | 0 | 0 | 100 | 0 | 0 | 0 | 100 | 0 |
| S5 | 0.3 | 594 | 23 | 28 | 0 | 72 | 0 | 0 | 100 | 0 | 0 | 0 | 100 | 0 |
| S6 | 0.3 | 557 | 25 | 52 | 0 | 47 | 0 | 1 | 100 | 0 | 0 | 0 | 100 | 0 |
| S7 | 0.7 | 564 | 25 | 44 | 0 | 56 | 0 | 0 | 99 | 0 | 0 | 0 | 100 | 0 |
| S8 | 1.1 | 404 | 29 | 32 | 0 | 68 | 0 | 0 | 1 | 0 | 99 | 0 | 100 | 0 |
| S9 | 6.6 | 490 | 33 | 36 | 1 | 62 | 0 | 0 | 14 | 0 | 86 | 0 | 99 | 1 |
| S10 | 10.0 | 475 | 33 | 50 | 3 | 47 | 0 | 0 | 38 | 6 | 53 | 3 | 14 | 86 |
| S11 | 38.0 | 445 | 35 | 40 | 2 | 58 | 0 | 0 | 24 | 1 | 73 | 1 | 45 | 55 |
| S12 | 0.2 | 496 | 45 | 39 | 0 | 60 | 0 | 0 | 0 | 0 | 100 | 0 | 0 | 100 |
| S13 | 8.6 | 393 | 31 | 29 | 7 | 54 | 10 | 0 | 6 | 18 | 58 | 14 | 0 | 100 |
| S14 | 0.5 | 220 | 24 | 35 | 4 | 61 | 0 | 0 | 0 | 5 | 95 | 0 | 0 | 100 |
| S15 | 83.2 | 386 | 31 | 32 | 2 | 65 | 1 | 0 | 19 | 5 | 72 | 3 | 20 | 80 |
| S16 | 3.0 | 308 | 23 | 16 | 0 | 84 | 0 | 0 | 30 | 28 | 42 | 0 | 0 | 100 |
| S17 | 5.0 | 340 | 24 | 29 | 2 | 69 | 0 | 0 | 17 | 41 | 42 | 0 | 0 | 100 |
| S18 | 21.3 | 342 | 34 | 17 | 2 | 81 | 0 | 0 | 6 | 0 | 94 | 0 | 0 | 100 |
| S19 | 143.4 | 351 | 32 | 28 | 2 | 69 | 1 | 0 | 13 | 7 | 78 | 2 | 12 | 88 |
| S20 | 159.0 | 339 | 32 | 28 | 2 | 69 | 1 | 0 | 11 | 7 | 79 | 2 | 11 | 89 |

[a]NF=natural forest, MF=managed forest, AM=agricultural mosaic (includes crops and pasture for cattle grazing), WB=water body, BG=bare ground.
[b]MOL=Mollisols, ENT= Entisols, INC=Inceptisols, MIS= Miscellaneous.
[c]PR=Playa Rica Formation, VI=Viche Formation.

**3.2 Hydrological data collection and analysis**

Water level (or stage) data were continuously recorded using HOBO® U20L pressure transducer water level loggers (USA; accuracy 4 mm) every 15 minutes at four sampling sites during the study period (January–December 2021) to identify the hydrological dynamics of the hydrological system. The sites were selected based on their hydrological regime. That is, the loggers were placed in two subcatchments presenting intermittent hydrological conditions (S7 and S8 in Fig. 1a) and in two

subcatchments with perennial flow conditions (S10 and S11). Water level hydrographs were normalized to a maximum value of 1 by dividing the observed water level data by the maximum water level recorded during the study period to ease visualization. Normalized water level data were used to characterize the temporal variability of the region's hydrological regime (e.g. Jachens et al., 2020; Kirchner et al., 2020) and identify differences in flow dynamics between intermittent and perennial streams using the Water Engineering Time Series PROcessing tool (WETSPRO; Willems, 2014). WETSPRO was



used to separate the hydrographs into baseflow, shallow subsurface flow (or interflow), and overland flow components and
to identify independent hydrological events (e.g. Correa et al., 2016; Lazo et al., 2019). The baseflow, shallow subsurface,
and overland flow contributions to total streamflow; the recession time of baseflow and shallow subsurface flow; and the
time to peak and time from peak to baseflow indices of event hydrographs were used to compare flow dynamics between
intermittent and perennials streams (Dingman, 2015). To identify potential differences in the hydrological dynamic of

intermittent streams during the wet and dry periods, we analysed the hydrological response of both types of streams during
each period.

### 3.3 Isotopic data collection and analysis

The isotopic composition of streamflow at the 20 sampling sites (S1-S20; Fig. 1a) was monitored to identify potential
differences in the hydrological behaviour of intermittent and perennial sites. Samples for isotopic analysis were collected

during six monitoring campaigns (M1-M6) carried out in the period January–December 2021 at each sampling site. The
campaigns were conducted roughly every two months, starting in late February 2021 (M1) and finalizing in mid-December
2021 (M6), to identify how the hydrological behaviour of the system varies during wet, transition, and dry periods.
Grab samples of stream water were filtered in situ using 0.45 μm polypropylene single-use syringe membrane filters
(Puradisc 25 PP Whatman Inc., USA) and stored in 2 ml amber glass vials. The vials were sealed with parafilm and

refrigerated at 10 °C to avoid the effects of evaporative fractionation. The oxygen-18 and hydrogen-2 isotopic ratios of the
water samples were measured using a cavity ringdown spectrometer (Picarro 2130-i) at the Water and Soil Quality
Laboratory of the University of Cuenca. Three Picarro secondary standards were used in the analysis: ZERO ($\delta^2$H =
0.3±0.2‰, $\delta^{18}$O = 1.8±0.9‰), MID ($\delta^2$H = -20.6±0.2‰, $\delta^{18}$O = -159.0±1.3‰), and DEPL ($\delta^2$H = -29.3±0.2‰, $\delta^{18}$O = -
235.0±1.8‰), obtaining very strong linear correlation in the analysis ($r^2$>0.99). The long-term analytical precision of the

instrument is 0.5‰ for hydrogen-2 ($\delta^2$H) and 0.1‰ for oxygen ($\delta^{18}$O). Six sample injections were applied and the first three
were discarded to avoid memory effects (Penna et al., 2012). For the last three injections, we calculated the maximum $\delta^{18}$O
and $\delta^2$H differences and compared them with the analytical precision of the instrument. The differences for all samples were
smaller than the analytical precision. The samples were checked for organic contamination using ChemCorrect™ (Picarro
Inc., USA), and no organic contamination was detected. The isotopic composition of the isotopic ratios is reported in per

mill values (‰) using the δ notation according to the Vienna Standard Mean Ocean Water (V-SMOW; (Craig, 1961). The
spatial (across the 20 sampling sites, S1-S20) and temporal (across the six monitoring campaigns, M1-M6) variability of the
stable isotopic composition of oxygen-18 ($\delta^{18}$O), hydrogen-2 ($\delta^2$H), and deuterium excess (*d-excess* = $\delta^2$H – 8·$\delta^{18}$O;
(Dansgaard, 1964) were used to identify differences in the hydrological behaviour of the system (e.g. Lahuatte et al., 2022;
Mosquera et al., 2016a).

Since no rainfall oxygen-18 and hydrogen-2 isotopic ratios were available in this study, we used data from nearby stations
belonging to the Global Network of Isotopes in Precipitation (GNIP; IAEA/WMO, 2021) as reference. We identified two
nearby stations possessing at least one year of monthly isotopic data. The Esmeraldas (30 m a.s.l.) and La Concordia (360 m





a.s.l.) GNIP stations are situated 66 and 52 km from the study site, respectively. The stations' metadata, including geographical information and summary statistics of the available data, are presented in Table S1 as online supplementary

material. Using those data, we constructed a Regional Meteoric Water Line (RMWL; (Craig, 1961), i.e. the linear relation between the $\delta^{18}O$ and $\delta^2H$ isotopic composition of the available rainfall isotopic composition in the region. We compared the isotopic composition of the stream water samples collected across the Cube River catchment to the RMWL to identify the potential influence of regional rainfall on the hydrological behaviour of the catchment.

### 3.4 Geochemical data collection and analyses

The geochemical composition of stream water samples was also monitored at the 20 sampling sites during the six monitoring campaigns in which samples for isotopic analysis were collected. The geochemical data were used to identify the potential effect of biophysical landscape features on the water flow paths influencing streamflow generation in intermittent and perennial streams across the Cube River catchment.

Geochemical monitoring included the in-situ measurement of water physicochemical parameters and the collection of stream
water samples to determine their solute composition via laboratory analyses. The physicochemical parameters included water temperature, electrical conductivity, dissolved oxygen, and pH. These parameters were measured in-stream at each sampling site during each monitoring campaign using a portable multiparameter probe (YSI, PRO DSS®; USA). The probe was laboratory-calibrated prior to each of the sampling campaigns.

To determine stream water dissolved solute concentrations, two types of samples were collected. Water samples for the
analysis of alkalinity, total N (TN), total P (P), $SO_4^=$, $NO_3^-$, $F^-$, and total organic carbon (TOC) were collected in two unfiltered 500 ml high-density polyethylene (HDPE; Thermo Scientific™ Nalgene™, USA) containers previously rinsed with 10% HCl. Alkalinity was measured according to the Standard Methods (SM) protocol SM 2320 B (APHA, 2012). TN concentrations were determined using the Kjeldahl method (Buchi, speed digester K436, scrubber K415 and KjelFlex K360) following the protocol SM 4500-Norg B (APHA, 2012). P was measured following the colorimetric analysis described in the
SM 4500-P B (APHA, 2012). $SO_4^=$, $NO_3^-$, and $F^-$ were analysed by the colorimetric methods SM 246 C, SM 4500 $NO_3^-$ D and SM 4500 $F^-$ C, respectively (APHA, 2012). TOC was determined by combustion catalytic oxidation using a Shimadzu TOC-L analyser, following the EPA 9060a method.

Water samples for analysing 18 dissolved elements, including Al, As, Ba, Ca, Cd, Co, Cu, Cr, Fe, K, Mg, Mn, Mo, Na, Ni, Pb, V, and Zn were collected in two 60 ml HDPE (Thermo Scientific™ Nalgene™, USA) bottles rinsed with 10% HCl. The
latter were filtered using 0.45 μm polypropylene single-use syringe membrane filters (Puradisc 25 PP Whatman Inc., USA) and preserved with 2% $HNO_3$. These elements were analysed with an ICP-OES (Thermo Scientific iCap 7000), following the Standard Method SM 3120 B.

In all analyses, the calibration curves' Pearson coefficient (r) was greater than 0.995 to guarantee accuracy. The method's detection limits (DL) were calculated as three times the standard deviation of the blank (EURACHEM/CITAC, 2012). All
the results obtained were the average of three repeated analyses. Quality assurance and quality control analyses were





performed every 10 samples applying standard dilutions for metals (ERA 500 Trace Metal certified reference material, Waters$^{TM}$), for TOC (1000 mg/L, ERA Waters$^{TM}$), and for TN (glycine, Across Organics). The geochemical analyses were conducted at the Core Lab de Ciencias Ambientales at Universidad San Francisco de Quito. Out of the 24 measured solutes, we only report the concentrations of elements that presented values above detection limits (such limits are presented in Table

S2 as supplementary material) at each sampling site and monitoring campaign to allow for a robust comparison of the spatiotemporal geochemical conditions of stream water across the nested monitoring system. The measured values of the reported solutes represented the average of the two replicates collected at each study site and monitoring campaign.

Relationships between pairs of solutes were examined through Spearman correlation (r) analysis using the median concentration of all samples collected during all monitoring campaigns at each sampling site. Spearman correlations were

considered statistically significant when presenting a 95% level of confidence (p < 0.05). If statistically significant, correlations were considered strong when r>0.75 and moderate when 0.50<r≤0.75 (Akoglu, 2018). Correlations were considered weak for r≤0.50, regardless of their statistical significance. The same statistical test was used to identify potential relations between the median solutes' concentrations and biophysical landscape characteristics (i.e. topography, soils, geology, and land cover).

**4 Results**

**4.1 Hydrological dynamics**

Normalized water level hydrographs of intermittent and perennial streams monitored across the Cube River catchment during the study period (January to December 2021) are shown in Fig. 2. The yearly hydrographs (Figs. 2a-2b) clearly depict the strong climate seasonality of the study area, with a wet period lasting from January to mid-June showing the frequent

occurrence of events generating the highest peaks throughout the year. Following the wet period, a transition period from mid-June to mid-July is observed. This period was characterized by the occurrence of events with less frequency that produced lower peak values than in the wet period. A dry period from mid-July to December subsequently occurred, in which the water level was generally the lowest except for a few events observed during late September-early October and late December 2021.

At a yearly time scale, the hydrographs show that intermittent streams (Fig. 2a) returned to baseflow faster after rainfall cessation than perennial streams, which presented longer recessions (Fig. 2b). As a result, the hydrograph of intermittent streams tended to return to a value of zero-flow more frequently than the hydrograph of perennial streams. The separation of the hydrographs into their subcomponents quantitatively depicts clear differences in the hydrological dynamic of both types of flow regimes. Baseflow was lower (0.20) and overland flow (0.15) was higher in intermittent streams (Fig. 2a) than in

perennial streams in which baseflow was 0.40 and overland flow was 0.05 (Fig. 2b). In addition, the recession times of baseflow (12.5 days) and shallow subsurface flow (0.4 days) of intermittent streams were shorter than in perennial streams, in which the recession times of baseflow and shallow subsurface flow were 18.8 and 1.7 days, respectively.




The hydrographs of representative events during the wet period in intermittent and perennial streams are presented in Fig. 2c and Fig. 2d, respectively. These event hydrographs show that the time to peak (2 h) and the time from peak to baseflow (60 h) in intermittent streams (Fig. 2c) were shorter than those in perennial streams in which the former was 3 h and the latter >89 h (Fig. 2d). Even though the time to peak and time from peak to baseflow were longer in events observed during the dry period at both types of streams (Figs. 2e-2f) compared to events occurring in the wet period, the difference in the dynamic between them was similar in both periods. That is, during the dry period the time to peak (5 h) and the time from peak to baseflow (87 h) in intermittent streams (Fig. 2e) were also lower than those in intermittent streams, which were 9 and 158 h (Fig. 2f), respectively.

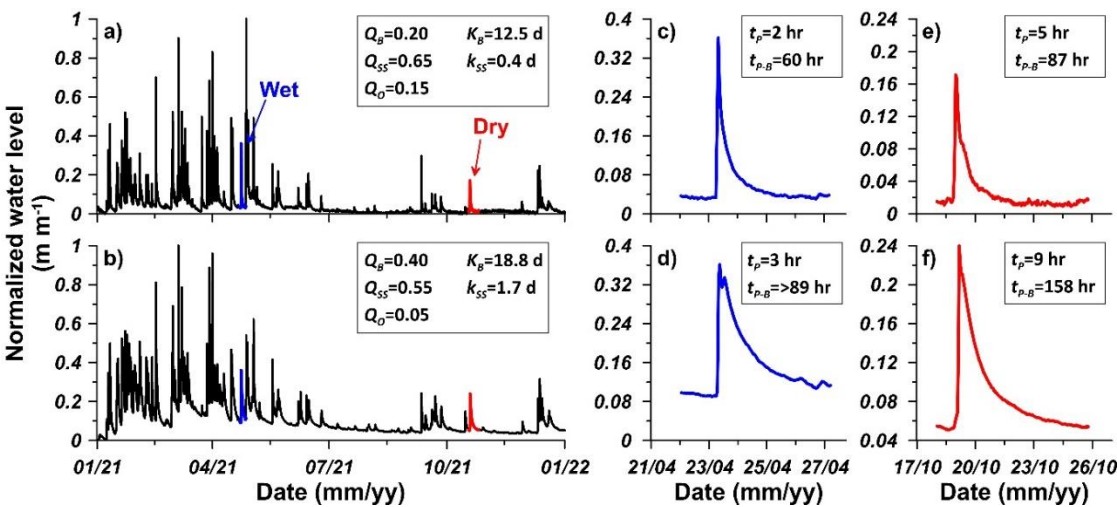

Figure 2: Hourly time series of normalized water level hydrographs showing the hydrological behavior of a) intermittent (sampling site S7) and b) perennial (sampling site S11) streams within the Cube River catchment during the period January-December 2021. Normalized water level hydrographs of representative events at c) intermittent and d) perennial sites during the wet period (shown in blue in subplots a and b). Normalized water level hydrographs of representative events at e) intermittent and f) perennial sites during the dry period (shown in red in subplots a and b). Abbreviations: $Q_B$=baseflow contribution to total streamflow; $Q_{SS}$= shallow subsurface flow or interflow contribution to total streamflow; $Q_O$=overland flow contribution to total streamflow; $k_B$=recession time of baseflow; $k_{SS}$=recession time of shallow subsurface flow or interflow; $t_P$ = time to peak; $t_{P-B}$ = time from peak to baseflow.

**4.2 Isotopic characterization of stream water**

Based on the hydrological analyses, the isotopic and geochemical data were collected during the wet (M1 and M2), transition (M3), and dry (M4-M6) periods. Figure 3 depicts that the isotopic composition of stream water remains relatively stable throughout the year, varying in a narrow isotopic range (-2.9 to -4.3‰ in $\delta^{18}O$ and -14.5.to -21.3‰ in $\delta^2H$); except for monitoring campaign M2 carried out in late April 2021 during the wettest period of the year (Figs. 2a-2b). During the wettest period, most sites showed lower isotopic values (i.e. more negative) than the rest of the year, reaching values as low as -4.9‰ in $\delta^{18}O$ and -27.6‰ in $\delta^2H$. Except for a few stream water samples mainly collected during the transition and dry periods, most stream water samples plotted above both the RMWL ($\delta^2H$ = 7.5·$\delta^{18}O$+8.1) and GMWL ($\delta^2H$ = 8·$\delta^{18}O$+10) (Fig. 3).





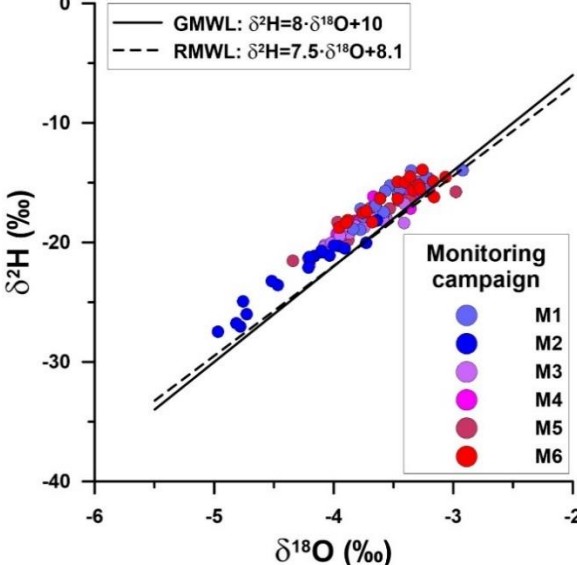


**Figure 3: Relationship between oxygen-18 ($\delta^{18}$O) and hydrogen-2 ($\delta^2$ H) isotopic ratios in stream water samples collected at the 20 sampling sites within the Cube River catchment during six monitoring campaigns (M1-M6) carried out during the wet (M1 and M2), transition (M3) and dry (M4, M5, and M6) periods in 2021. Monitoring campaigns M2 and M6 correspond to the wettest (dark blue circles) and driest (red circles) sampling periods, respectively. The Global Meteoric Water Line (GMWL) and a**
**Regional Meteoric Water Line (RMWL) constructed using available data from two nearby stations of the Global Network of Isotopes in Precipitation (GNIP) database** (IAEA/WMO, 2021) **are also displayed for reference. Metadata, geographical information, isotopic data statistics, and source of the GNIP stations used in this analysis are presented in Table S1 as supplementary material.**

During the wettest period, the isotopic composition of $\delta^{18}$O in the smallest subcatchments (S1-S8) was depleted
(mean±standard deviation: -4.7±0.3‰) compared to the rest of the sampling sites possessing larger drainage areas (-4.1±0.3‰) (Fig. 4a). Differently, during the dry period no systematic difference in the $\delta^{18}$O isotopic composition across the sampling sites was identified (-3.5±0.3‰), and most sites generally presented isotopic values close to the average value (-3.7‰) of all samples collected during all monitoring campaigns (Fig. 4b). During the rest of the monitoring campaigns carried out during the beginning of the wet period (M1), the transition period (M3), and the rest of the dry period (M4 and
M5), the isotopic composition of $\delta^{18}$O across the catchments was similar to that of the driest period shown in Fig. 4b (Fig. S1 in supplementary material). The isotopic composition of $\delta^2$H presented similar spatial temporal patterns than those shown for $\delta^{18}$O and thus is not shown for brevity. Regarding *d-excess*, this parameter plotted consistently close to the average of all samples collected during all monitoring campaigns (+11.6‰) regardless of the monitoring period (Figs. 4c-4d), showing no spatial and temporal differences among sampling sites and monitoring periods.





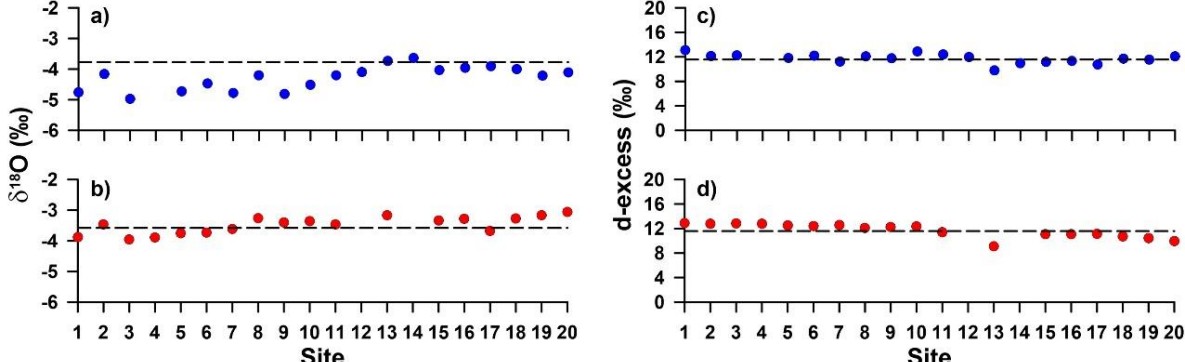

**Figure 4: (a, b) Oxygen-18 (δ¹⁸O) isotopic ratios and (c, d) *d-excess* values in stream water samples collected at the 20 sampling sites within the Cube River catchment during the wettest (dark blue circles in a and c) and driest (red circles in b and d) sampling periods (i.e. M2 and M6 in Fig. 3). The dashed lines in a) and b) represent the average δ¹⁸O value (-3.7‰) and in c) and d) represent the average *d-excess* value (+11.6%) of the samples collected at all monitoring sites during the six monitoring campaigns carried out in 2021 for reference, respectively. In the x-axis, the sampling sites are ordered according to the elevation of their outlets, generally corresponding to the sites with the smallest drainage areas to be located at the Cube River headwaters and the sites with the largest drainage areas downstream toward the catchment's outlet (Fig. 1a and Table 1).**

### 4.3 Geochemical characterization of stream water

It is worth noting that solutes that cause harmful effects on humans and aquatic ecosystems' health even at small concentrations (WHO Water, 2008), such as As, Cd, Co, Cu, Cr, and Ni, were not detected in any of the analysed samples. Other elements that presented concentrations below detection limits for more than 20% (n=4) of the sampling sites for at least one monitoring campaign were not included in the analyses (i.e. TN, NO$_3^-$, Al, Fe, Mo, V, and Zn). Therefore, in total 11 solutes (i.e. SO$_4^=$, F$^-$, P, TOC, Ba, Ca, K, Mg, Mn, Na, and Pb) in addition to alkalinity, chemical oxygen demand (COD), and the in-situ measured physicochemical parameters of water (i.e. temperature, electrical conductivity, dissolved oxygen, and pH) are reported (i.e. 17 geochemical parameters).

Table 2 depicts a wide range of variations in the water geochemical characteristics along the Cube River catchment. Except for P, TOC, and Pb, the lowest values of the geochemical parameters were generally observed in the headwaters of the Cube River catchment (S1-S7). In contrast, the sampling sites with the largest drainage areas (S17-S20) tended to present the largest values. Even though the largest values of P and TOC were observed at the headwater catchment S2; the lowest value for P was found at S3 and S6, and the lowest value for TOC was observed at S12 (Table 2). Pb displayed the lowest value at S20 and the largest value at S12.

The Spearman statistical analysis showed strong positive correlations among several pairs of stream water physical-chemical parameters. These parameters included Ca, electrical conductivity, alkalinity, SO$_4^=$, F$^-$, Na, Mg, and temperature (Fig. 5). The aforementioned parameters and DO, pH, P, Pb, COD, and Ba were moderately, positively, and significantly correlated. Mn showed a moderate and positive correlation with COD, and a weak and negative correlation with DO. TOC only showed a moderate and negative correlation with Ba.





**Table 2: Median and standard deviation (Sd) of stream water physical-chemical parameters and solutes' concentrations of the 20 sampling sites monitored within the Cube River catchment. All sites were sampled six times in 2021, except for site S14 (marked with * symbol) that completely dried during three sampling campaigns in the dry season. Electrical conductivity (EC) and temperature (T) are reported in µS cm$^{-1}$ and °C, respectively. Solutes' concentrations are reported in ppm, except for Ba, Pb, and Mn that are reported in ppb. Green and red shading indicates the sites where the lowest and highest values of the monitored water chemical parameters were observed. EC=electrical conductivity, Alk=Alkalinity, T=Temperature, DO=dissolved oxygen, COD=chemical oxygen demand, and TOC=total organic carbon.**

| Site | Statistic | K | EC | Ca | Alk | SO$_4$ | Na | Mg | F | T | DO | P | Pb | COD | Ba | pH | Mn | TOC |
|---|---|---|---|---|---|---|---|---|---|---|---|---|---|---|---|---|---|---|
| S1 | Median | 3.3 | 132 | 11.0 | 44.8 | 24.0 | 5.6 | 3.91 | 0.05 | 21.6 | 7.6 | 0.28 | 5.4 | 13.4 | 61.0 | 7.3 | 8.7 | 2.7 |
| | Sd | 1.3 | 45 | 4.7 | 13.1 | 9.8 | 2.0 | 1.64 | 0.02 | 0.6 | 0.8 | 0.06 | 0.6 | 6.5 | 39.3 | 0.4 | 7.4 | 1.8 |
| S2 | Median | 2.8 | 146 | 12.1 | 52.7 | 23.5 | 6.3 | 4.24 | 0.06 | 22.1 | 8.8 | 0.49 | 5.8 | 17.9 | 54.4 | 7.8 | 2.3 | 2.8 |
| | Sd | 0.8 | 48 | 4.5 | 18.7 | 5.4 | 2.4 | 1.49 | 0.02 | 0.4 | 0.2 | 0.17 | 2.1 | 27.3 | 35.5 | 0.4 | 1.1 | 1.3 |
| S3 | Median | 1.2 | 69 | 3.9 | 20.4 | 10.1 | 4.3 | 1.77 | 0.04 | 21.8 | 5.0 | 0.27 | 6.0 | 17.3 | 40.7 | 6.9 | 13.2 | 1.8 |
| | Sd | 0.5 | 20 | 1.3 | 4.7 | 3.4 | 1.2 | 0.55 | 0.01 | 0.5 | 1.8 | 0.13 | 1.2 | 10.9 | 22.0 | 0.4 | 8.8 | 1.2 |
| S4 | Median | 2.5 | 120 | 11.8 | 49.4 | 8.4 | 5.3 | 3.32 | 0.05 | 22.4 | 7.2 | 0.29 | 6.7 | 26.2 | 40.9 | 7.5 | 20.1 | 1.9 |
| | Sd | 0.6 | 46 | 5.0 | 18.8 | 3.4 | 2.0 | 1.28 | 0.02 | 0.9 | 0.8 | 0.18 | 0.7 | 9.2 | 20.4 | 0.6 | 8.0 | 1.0 |
| S5 | Median | 2.5 | 114 | 9.7 | 44.0 | 17.2 | 5.7 | 3.24 | 0.06 | 21.5 | 8.4 | 0.31 | 5.4 | 14.5 | 43.5 | 7.6 | 0.8 | 2.0 |
| | Sd | 0.8 | 36 | 3.6 | 15.0 | 3.0 | 1.7 | 1.14 | 0.02 | 0.8 | 0.1 | 0.13 | 1.4 | 6.9 | 26.7 | 0.3 | 3.1 | 0.5 |
| S6 | Median | 2.1 | 108 | 8.5 | 34.0 | 16.6 | 5.9 | 3.06 | 0.06 | 21.7 | 8.6 | 0.27 | 4.8 | 16.8 | 50.8 | 7.8 | 7.2 | 1.7 |
| | Sd | 0.6 | 30 | 2.5 | 9.4 | 4.5 | 1.6 | 0.98 | 0.01 | 0.8 | 0.5 | 0.40 | 2.1 | 6.0 | 35.3 | 0.5 | 5.9 | 0.5 |
| S7 | Median | 2.7 | 131 | 9.7 | 41.4 | 17.9 | 8.2 | 3.57 | 0.06 | 21.5 | 7.8 | 0.30 | 4.9 | 20.2 | 58.8 | 7.5 | 8.3 | 1.8 |
| | Sd | 0.9 | 48 | 3.5 | 11.2 | 4.6 | 4.1 | 1.28 | 0.01 | 0.5 | 0.6 | 0.10 | 1.9 | 8.7 | 35.7 | 0.2 | 33.5 | 0.2 |
| S8 | Median | 4.4 | 264 | 20.2 | 81.2 | 41.4 | 14.6 | 7.00 | 0.08 | 22.5 | 9.0 | 0.33 | 5.6 | 16.8 | 100.2 | 8.0 | 1.9 | 1.8 |
| | Sd | 1.8 | 112 | 9.1 | 33.8 | 15.8 | 8.0 | 3.08 | 0.02 | 0.8 | 0.2 | 0.16 | 2.3 | 6.5 | 58.4 | 0.3 | 0.5 | 0.6 |
| S9 | Median | 3.3 | 181 | 14.0 | 60.4 | 29.0 | 9.1 | 5.26 | 0.07 | 23.1 | 8.9 | 0.39 | 5.3 | 20.0 | 67.0 | 7.8 | 1.7 | 1.9 |
| | Sd | 1.1 | 71 | 5.6 | 21.5 | 9.5 | 4.5 | 2.04 | 0.02 | 0.7 | 0.5 | 0.17 | 1.6 | 6.0 | 46.0 | 0.5 | 3.7 | 1.7 |
| S10 | Median | 3.5 | 219 | 22.5 | 51.4 | 30.9 | 7.3 | 5.07 | 0.07 | 23.5 | 9.9 | 0.39 | 5.7 | 18.4 | 57.0 | 8.2 | 3.3 | 1.7 |
| | Sd | 1.1 | 78 | 8.3 | 22.3 | 9.6 | 2.9 | 1.81 | 0.02 | 0.6 | 0.8 | 0.14 | 2.4 | 38.0 | 30.9 | 0.6 | 3.7 | 0.9 |
| S11 | Median | 3.8 | 223 | 20.4 | 69.6 | 34.7 | 9.8 | 5.52 | 0.08 | 24.2 | 8.8 | 0.35 | 5.6 | 22.2 | 63.4 | 8.2 | 4.5 | 2.0 |
| | Sd | 1.1 | 67 | 6.2 | 21.3 | 8.2 | 3.5 | 1.60 | 0.03 | 0.9 | 0.5 | 0.16 | 2.7 | 6.8 | 19.8 | 0.3 | 1.8 | 1.2 |
| S12 | Median | 5.3 | 275 | 24.1 | 91.1 | 29.3 | 13.8 | 8.54 | 0.09 | 23.0 | 6.9 | 0.44 | 9.3 | 24.1 | 109.2 | 7.4 | 35.0 | 1.1 |
| | Sd | 0.9 | 29 | 3.4 | 15.4 | 6.9 | 3.5 | 1.22 | 0.02 | 0.7 | 0.8 | 0.19 | 4.8 | 15.0 | 17.6 | 0.2 | 17.7 | 0.3 |
| S13 | Median | 3.0 | 132 | 11.7 | 53.3 | 10.1 | 6.4 | 3.49 | 0.07 | 24.0 | 7.9 | 0.46 | 6.8 | 31.8 | 46.4 | 7.6 | 21.1 | 2.6 |
| | Sd | 0.6 | 32 | 3.3 | 11.0 | 2.7 | 1.6 | 0.96 | 0.02 | 1.2 | 0.3 | 0.19 | 2.0 | 12.0 | 24.6 | 0.3 | 9.0 | 0.7 |
| S14* | Median | 6.5 | 488 | 56.1 | 94.3 | 151.5 | 13.9 | 8.02 | 0.14 | 26.1 | 8.0 | 0.48 | 8.0 | 25.8 | 183.1 | 7.7 | 35.6 | 1.5 |
| | Sd | 1.6 | 100 | 9.8 | 21.0 | 139.5 | 5.3 | 1.51 | 0.02 | 0.6 | 0.9 | 0.23 | 2.1 | 4.1 | 44.6 | 0.0 | 24.0 | 0.6 |
| S15 | Median | 4.8 | 356 | 34.4 | 81.0 | 63.4 | 15.4 | 7.72 | 0.12 | 25.9 | 9.2 | 0.42 | 5.6 | 25.8 | 69.1 | 8.0 | 20.1 | 2.0 |
| | Sd | 0.9 | 79 | 5.9 | 20.7 | 39.4 | 3.7 | 1.19 | 0.03 | 0.9 | 0.7 | 0.21 | 3.4 | 5.5 | 24.9 | 0.4 | 9.4 | 0.5 |
| S16 | Median | 4.5 | 391 | 48.3 | 92.1 | 74.0 | 9.8 | 6.92 | 0.09 | 24.5 | 8.7 | 0.48 | 5.5 | 28.8 | 97.2 | 8.3 | 7.6 | 2.0 |
| | Sd | 1.5 | 135 | 17.5 | 29.6 | 35.2 | 3.6 | 2.28 | 0.02 | 0.8 | 1.0 | 0.20 | 3.5 | 7.8 | 30.4 | 0.6 | 4.3 | 1.0 |
| S17 | Median | 6.3 | 500 | 50.9 | 82.7 | 163.6 | 19.5 | 9.37 | 0.17 | 24.3 | 8.2 | 0.43 | 5.1 | 32.0 | 106.1 | 7.8 | 5.6 | 1.6 |
| | Sd | 1.6 | 136 | 13.9 | 26.0 | 81.0 | 7.5 | 2.04 | 0.04 | 0.5 | 0.7 | 0.22 | 4.2 | 14.2 | 22.5 | 0.3 | 32.3 | 0.6 |
| S18 | Median | 8.0 | 519 | 54.9 | 93.5 | 108.2 | 22.3 | 8.11 | 0.13 | 25.3 | 9.7 | 0.43 | 5.3 | 23.8 | 78.8 | 8.6 | 1.7 | 2.1 |
| | Sd | 2.7 | 142 | 14.3 | 29.7 | 44.8 | 11.4 | 2.12 | 0.03 | 0.8 | 1.4 | 0.25 | 3.3 | 14.0 | 26.0 | 0.5 | 4.6 | 0.7 |
| S19 | Median | 7.1 | 493 | 51.6 | 101.6 | 81.4 | 21.9 | 9.03 | 0.14 | 26.6 | 10.0 | 0.40 | 5.7 | 23.2 | 86.1 | 8.4 | 6.3 | 1.8 |
| | Sd | 1.6 | 81 | 7.9 | 15.4 | 121.3 | 6.0 | 0.74 | 0.02 | 0.7 | 2.2 | 0.31 | 5.5 | 7.7 | 17.1 | 0.4 | 2.1 | 0.7 |
| S20 | Median | 6.8 | 517 | 53.2 | 118.2 | 88.5 | 19.0 | 8.88 | 0.14 | 26.0 | 9.4 | 0.46 | 4.1 | 27.5 | 90.7 | 8.2 | 16.0 | 1.8 |
| | Sd | 1.4 | 168 | 7.7 | 19.7 | 101.3 | 5.2 | 1.08 | 0.03 | 1.9 | 1.5 | 0.19 | 4.6 | 8.6 | 8.2 | 0.2 | 11.0 | 0.4 |



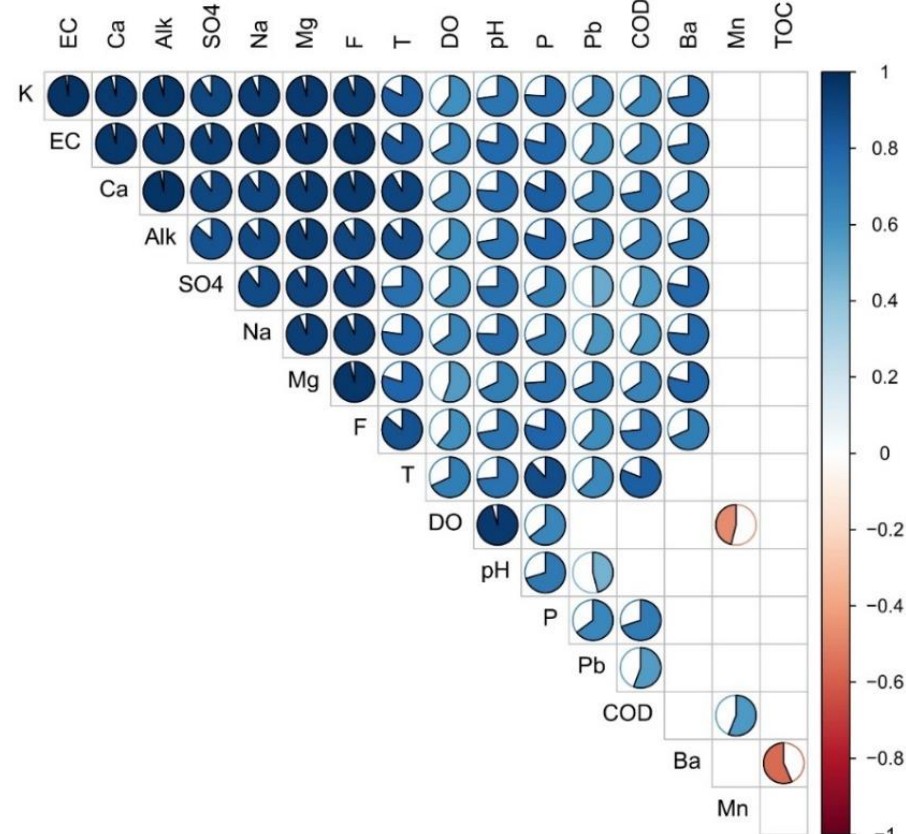

**Figure 5: Spearman correlation analysis between the medians of pairs of stream water physical-chemical parameters and solutes' concentrations of the 20 sampling sites monitored within the Cube River catchment shown in Table 3. Circles are shown only when the correlation is statistically significant (p-value <0.05). Colour intensity and the shaded area of each circle represent the absolute value of the corresponding correlation coefficient. Positive correlations are displayed in blue and negative correlations are shown in red. EC=electrical conductivity, Alk=Alkalinity, T=Temperature, DO=dissolved oxygen, COD=chemical oxygen demand, and TOC=total organic carbon.**


Considering the relatively large number of geochemical parameters analysed in this study and the aforementioned degree of correlation among them, in the following, we show and describe the spatial variability of water chemistry along the Cube River using three representative parameters of the geologic, biologic, and land use characteristics of the study area for brevity (e.g. Peña et al., 2023). Given the geogenic nature of Ca in dominating calcite veins in the Viche geological formation, its concentration was used to analyse how the geology from both formations (i.e. Playa Rica and Viche) affects






stream water chemistry. Because of the biogenic nature of TOC and the potential presence of organic matter in darkish lulita dominating the Playa Rica geological formation, TOC was used to assess the effects of forest degradation in the study area. In contrast, the concentration of P was used as an indicator of changes in land use due to other anthropogenic impacts

potentially affecting the hydrological behaviour and chemical characteristics of water at the study site including recreation, cultivation, and cattle grazing.

Ca concentrations varied between 3.9 and 54.9 ppm (Table 2). The lowest concentrations of Ca were observed in the smallest subcatchments located in the headwaters (S1-S7; Fig. 6a). The largest subcatchment and the catchment outlet (S14-S20) located at the lower part of the study area possessed the largest Ca concentrations. Between these extremes, medium

size subcatchments mainly located in the middle part of the catchment (S8-S13) have intermediate Ca concentrations. Concentrations of TOC ranged between 1.1 and 2.8 ppm (Table 2). The highest TOC concentrations are found at the sampling sites located in the headwaters of the catchment (S1-S7; Fig. 6b). In the middle and lower parts of the catchment TOC concentrations were generally smaller than in the headwaters. Although the variation of P across the sampling sites was relatively small, i.e. 0.27 and 0.49 ppm (Table 2), the concentration of P tended to increase from the headwaters to the outlet

of the catchment (Fig. 6b), except for subcatchment S2.

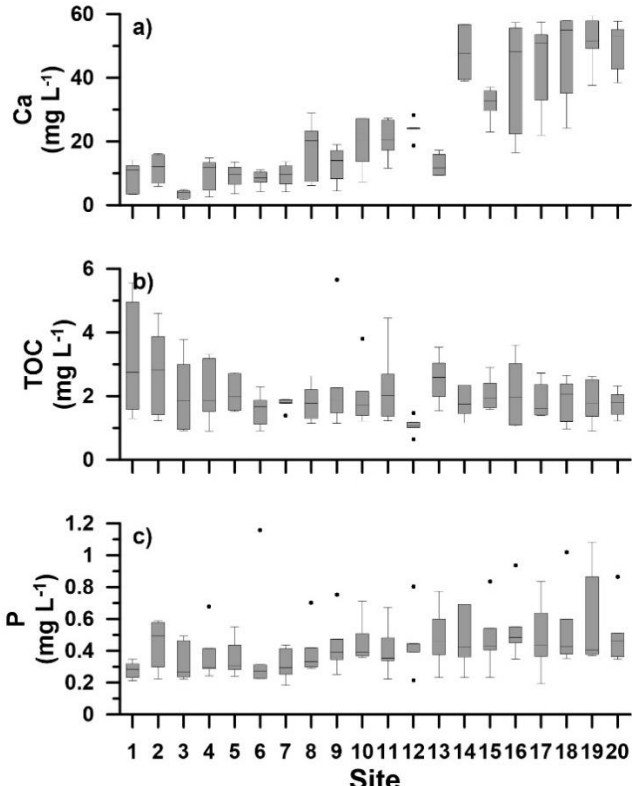

**Figure 6: Boxplots of the concentration of Calcium (Ca), total organic carbon (TOC), and phosphorus (P) across the 20 sampling sites monitored within the Cube River catchment during six monitoring campaigns carried out in 2021.**





## 4.4 Relationship between landscape features and stream water geochemistry

The assessment showed a relation between Ca, mean altitude, and geology (Table 3; Fig. 7). The 3D surface plot (Fig. 7a) depicts that: i) the Ca concentration decreases as mean altitude increases, ii) the concentration of Ca increases as the areal extent of the Viche formation increases, and iii) the areal extent of the Viche formation increases as mean altitude decreases. The correlation analysis shows a strong, negative, and significant correlation (r=-0.87) between Ca and mean altitude (Fig. 7b); a strong, positive, and significant correlation (r=0.78) between Ca and the Viche formation (Fig. 7c); and a strong, negative, and significant correlation (r=-0.79) between mean altitude and the Viche formation. Significant but weaker correlations were found between Ca and drainage area (r=0.69) and between Ca and Entisols (r=0.64). Weaker and/or non-statistically significant correlations were found between Ca and other landscape features (Table 3). Naturally, all correlations found between Ca and the landscape characteristics hold for other geochemical parameters strongly and moderately correlated with Ca (Fig. 5), including P. The correlation between Ca and the Playa Rica formation is the mirror image of that with the Viche formation shown in Fig. 7c. Weak and non-statistically significant correlations between TOC and landscape features were observed (Table 3).

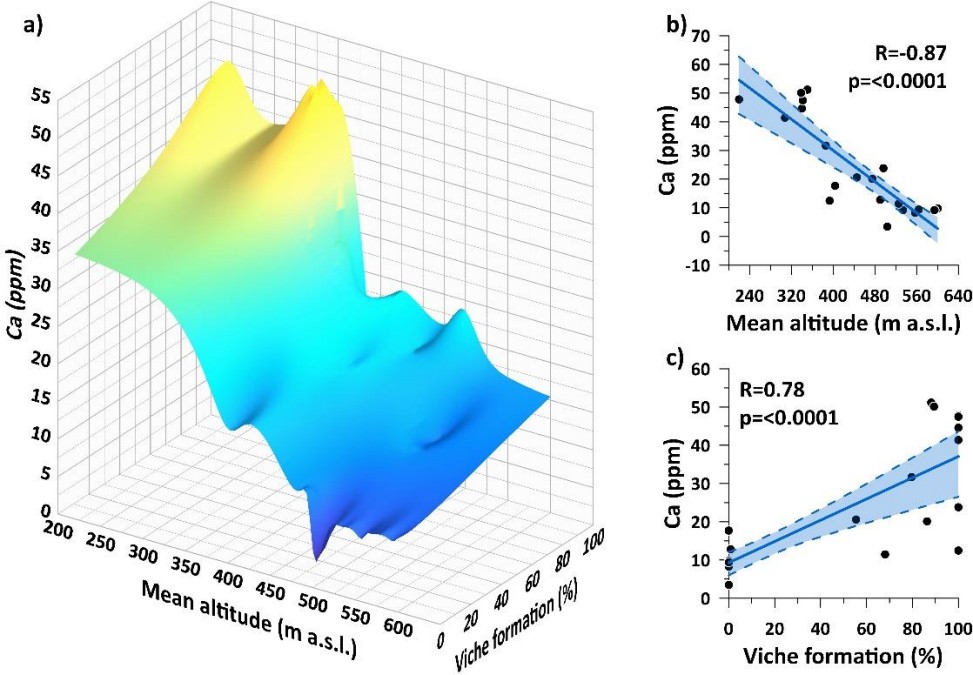

**Figure 7: a) 3D surface plot showing the relation among mean altitude, the areal extent of the Viche formation, and Ca concentration for the 20 sampling sites monitored within the Cube River catchment during six monitoring campaigns carried out in 2021. Correlations b) between mean altitude and Ca concentration and c) between the areal extent of the Viche formation and Ca concentration. Note that the color scheme in a) is only used to visualize the interplay between the predictor variables (mean altitude and Viche formation) and the response variable Ca concentrations.**





**Table 3: Spearman correlation coefficients of the relation between the medians of pairs of stream water physical-chemical parameters or solutes' concentrations and main landscape features of the 20 sampling sites monitored within the Cube River catchment. Blue and red shading indicates correlation coefficients > 0.75 and values in bold are statistically significant with a p<0.01. EC=electrical conductivity, T=Temperature, COD=chemical oxygen demand, Alk=Alkalinity, DO=dissolved oxygen, and TOC=total organic carbon.**

| Element | Drainage area (km²) | Mean altitude (m a.s.l.) | Mean slope (%) | Land cover[a] (%) | | | | | Distribution of soil types[b] (%) | | | | Geology[c] (%) | |
|---|---|---|---|---|---|---|---|---|---|---|---|---|---|---|
| | | | | NF | MF | AM | WB | BG | MOL | ENT | INC | MIS | PR | VI |
| Ba | 0.18 | -0.58 | 0.25 | -0.14 | 0.07 | 0.27 | -0.06 | -0.32 | -0.46 | 0.20 | 0.43 | -0.28 | -0.44 | 0.44 |
| Ca | 0.69 | **-0.87** | 0.48 | -0.53 | 0.38 | 0.54 | 0.47 | -0.41 | -0.28 | 0.64 | 0.23 | 0.29 | **-0.78** | **0.78** |
| K | 0.63 | **-0.83** | 0.50 | -0.47 | 0.44 | 0.49 | 0.45 | -0.37 | -0.41 | 0.53 | 0.37 | 0.19 | **-0.75** | **0.75** |
| Mg | 0.62 | **-0.80** | 0.50 | -0.38 | 0.32 | 0.42 | 0.38 | -0.36 | -0.36 | 0.56 | 0.31 | 0.22 | -0.74 | 0.74 |
| Mn | -0.17 | -0.19 | -0.15 | -0.06 | 0.22 | -0.06 | 0.10 | 0.09 | -0.28 | 0.31 | 0.08 | -0.02 | -0.38 | 0.38 |
| Na | 0.67 | **-0.78** | 0.53 | -0.40 | 0.30 | 0.47 | 0.41 | -0.35 | -0.30 | 0.43 | 0.27 | 0.13 | -0.64 | 0.64 |
| Pb | 0.44 | -0.50 | 0.33 | -0.54 | -0.08 | 0.53 | 0.42 | -0.43 | -0.16 | 0.47 | 0.11 | 0.33 | -0.56 | 0.56 |
| Alk | 0.63 | **-0.83** | 0.52 | -0.52 | 0.32 | 0.53 | 0.47 | -0.43 | -0.38 | 0.52 | 0.35 | 0.24 | **-0.76** | **0.76** |
| F | 0.69 | **-0.85** | 0.51 | -0.41 | 0.43 | 0.42 | 0.46 | -0.35 | -0.29 | 0.66 | 0.20 | 0.29 | **-0.79** | **0.79** |
| SO₄ | 0.55 | **-0.85** | 0.31 | -0.33 | 0.35 | 0.43 | 0.24 | -0.38 | -0.32 | 0.53 | 0.30 | 0.08 | -0.64 | 0.64 |
| COD | 0.48 | **-0.76** | 0.19 | -0.51 | 0.37 | 0.42 | 0.52 | -0.12 | -0.24 | **0.76** | 0.08 | 0.26 | -0.74 | 0.74 |
| P | 0.65 | **-0.81** | 0.41 | -0.55 | 0.32 | 0.44 | 0.53 | -0.22 | -0.16 | 0.72 | 0.02 | 0.39 | **-0.86** | **0.86** |
| TOC | 0.04 | 0.17 | -0.19 | 0.04 | 0.23 | -0.13 | 0.10 | 0.30 | 0.13 | 0.05 | -0.08 | 0.25 | 0.14 | -0.14 |
| T | **0.76** | **-0.87** | 0.48 | -0.53 | 0.43 | 0.44 | 0.67 | -0.22 | -0.26 | 0.72 | 0.19 | 0.49 | **-0.77** | **0.77** |
| pH | **0.84** | -0.70 | 0.51 | -0.43 | 0.24 | 0.47 | 0.49 | -0.20 | 0.06 | 0.52 | 0.01 | 0.37 | -0.48 | 0.48 |
| EC | 0.66 | **-0.86** | 0.48 | -0.44 | 0.41 | 0.47 | 0.41 | -0.38 | -0.34 | 0.54 | 0.30 | 0.20 | **-0.77** | **0.77** |
| DO | **0.85** | -0.56 | 0.54 | -0.32 | 0.25 | 0.34 | 0.49 | -0.12 | 0.14 | 0.45 | -0.03 | 0.49 | -0.34 | 0.34 |

**5 Discussion**

One of the key questions regarding the hydrology of intermittent hydrological systems is to determine how the interplay among hydrometeorological conditions, catchment surface and subsurface features, and land use influence streamflow dynamic and flow generation (Costigan et al., 2016; Godsey and Kirchner, 2014). Addressing this question is especially critical in tropical regions where water availability is drastically affected by changes in land use and climate. We present a
novel process-based comprehension of streamflow generation in a nested system of tropical intermittent and perennial streams. The results of this research conducted within the Chocó-Darien ecoregion provide a template for hydrological assessment in places where data is limited. In the following, we discuss the main results and propose a conceptual model of how water flow paths are shaped in these streams by the interaction of hydrometeorological, landscape, and land use conditions.

**5.1 Do hydrometeorological conditions influence streamflow generation in intermittent and perennial tropical streams?**

The isotopic composition of stream water allows assessing the influence of hydrometeorological conditions via evapotranspiration on the dynamics of intermittent and perennial streams within the Cube River catchment. Even though evapotranspiration was found to influence streamflow generation in an intermittent low relief and highly weathered
catchment with a subtropical climate in North Carolina, USA (Zimmer and McGlynn, 2017), the virtually negligible spatial and temporal variation of the *d-excess* isotopic fingerprint across our nested monitoring system throughout the year (Figs 4c,



4d) suggests that evapotranspiration has marginal or no influence on the hydrological behaviour of intermittent and perennial streams. This observation is likely explained by the high rainfall inputs during the wet period and the continuous cloudy and foggy conditions during the dry season observed at the study area. These hydrometeorological conditions likely reduce water

losses to the atmosphere via evapotranspiration (e.g. Mosquera et al., 2024; Ochoa-Sánchez et al., 2020). This finding is in line with those reported at montane ecosystems at higher elevations in north and south Ecuador, where high air humidity combined with cool to cold temperatures and/or rainy to foggy conditions lead to a diminished influence of evapotranspiration on stream water isotopic composition (Lahuatte et al., 2022; Mosquera et al., 2016a; Timbe et al., 2014). In addition, the fact that the isotopic composition of stream water across the sites monitored within the Cube River

catchment was higher than the RMWL built using rainfall isotopic data from nearby GNIP stations (Fig. 3) depicts that data from such stations is not representative of the isotopic composition of rainfall at the study site. This finding could be caused by the differences in altitude between the GNIP stations (30 and 360 m a.s.l.) and the study area with an elevation range of 53 and 688 m a.s.l. Another factor that could explain this result is the influence of the Mache-Chindul Mountain range in the hydrometeorological conditions of the study area, which likely does not affect or affects differently the isotopic composition

of rainfall at the GNIP stations located approximately 50 km away from the study site. These findings highlight the necessity to characterize the isotopic composition of rainfall at the Cube River catchment to understand further and quantify the influence of rainfall on flow generation processes through the analysis of the origin, age, and fate of water (e.g. Burt et al., 2023; Mosquera et al., 2016b).

**5.2 How do catchment subsurface features influence streamflow dynamics in intermittent and perennial tropical**
**streams?**

On the one hand, the flashy response of streamflow to rainfall inputs year-round (Figs. 2a,2c,2e) and the depletion of the isotopic composition of stream water in the headwaters of the Cube River catchment during the wettest period (Fig. 4a) suggest a strong influence of rainfall in water transport and tracer mixing mechanisms in subcatchments draining intermittent streams (e.g. Mosquera et al., 2020). On the other hand, these observations are supported by the low concentration of major

elements and electrical conductivity in those streams (Figs. 5,6a), indicating a low contact time of water with minerals in the subsurface. These findings are explained by the low bedrock permeability of the Playa Rica formation that dominates in intermittent subcatchments, which in turn reduces their subsurface water storage capacity. The latter does not only cause a fast response of streamflow to rainfall inputs but also its rapid deactivation as rainfall stops, causing the temporal cessation of flow during sustained periods of little to no rainfall.

In contrast, the lower influence of rainfall in streamflow dynamics of perennial streams is evidenced by the longer time spans necessary for those streams to reach peak flow and to return to base flow after rainfall events (Figs. 2b,2d,2f) compared to intermittent streams. The little to no depletion of the isotopic composition of stream water throughout the year (Fig. 4a,4b) and the higher concentration of major elements and electrical conductivity observed in perennial streams (Figs. 5,6a) support this finding and indicate a higher contact time of water with the underlying bedrock. The latter likely results from the high



permeability of the Viche formation that dominates in catchments draining perennial streams, providing them with a higher subsurface water storage capacity than catchments where intermittent flow governs (Lazo et al., 2019). The interplay between large inputs of rainfall during the wet period and the high subsurface water storage in perennial catchments explains their high flow regulation capacity, i.e. the sustained generation of flow throughout the year, including dry periods with little to no rainfall.

Other field studies focused on evaluating the role of surface and subsurface catchment conditions in the hydrological behaviour of intermittent hydrological systems have been previously conducted in environments with Mediterranean (Banda et al., 2023), semi-arid (Bourke et al., 2021), continental (Hatley et al., 2023), and tropical savannah (Farrick and Branfireun, 2015) climates. Those studies concluded that bedrock permeability plays a key role in streamflow activation and cessation. Our results in a highly seasonal tropical setting further support those findings, highlighting the key role catchment geological

features play in driving the occurrence of hydrological intermittency regardless of hydrometeorological condition. Opposed to these findings from field studies, numerical modelling approaches to investigate streamflow generation in intermittent hydrological systems to date have mainly focused on assessing the role of topography and/or soils (e.g. Gutiérrez-Jurado et al., 2019; Gutierrez-Jurado et al., 2021; Ilja Van Meerveld et al., 2019). Therefore, future modelling studies should carefully consider catchments' geological features to obtain a sound understanding of streamflow generation in intermittent

hydrological systems (Mimeau et al., 2024; Shanafield et al., 2021).

It is also worth noting that effective conservation and protection efforts currently target the Cube River catchment headwaters where the last plots of primary and secondary forest remain. Considering the importance of the middle and lower parts of the catchment whose geological features allow for continuous flow generation that maintains the consequent ecosystem services downstream, our findings suggest that conservation efforts should transcend into securing groundwater

replenishment and streamflow generation as part of future landscape conservation plans.

**5.3 How do changes in land use influence the hydrology of an intermittent system of tropical streams?**

Despite the large legacy of anthropogenic impacts in the Ecuadorian forest of the Chocó-Darien due to deforestation and cultivation (Fig. 1b; Molinero et al., 2019), we observed relatively little influence of land use change on the hydrological dynamics of intermittent and perennial streams in the Cube River catchment. The weak and non-statistically significant

correlation between geochemical tracers and the land cover features of the monitored sites (Table 3) suggests that land use had low effect on the water flow paths supporting streamflow generation as opposed to the influence of geology. Despite this general trend, higher concentrations of TOC at intermittent streams draining conserved forested areas situated at the upper part of the catchment (Fig. 6b) are likely explained by forest litter accumulation in the ground that is released to streams during rainfall events. On the contrary, the generally lower TOC concentrations found at the middle and lower parts of the

Cube River catchment likely result from the long-term legacy of forest loss to cultivation and cattle grazing that, in turn, affect stream water quality. These inferences are supported by higher concentrations of dissolved carbon from tropical





forests in comparison to other land cover including pastures for cattle grazing that have been reported in previous studies in tropical forests (e.g. Meyer et al., 1998; Moeller et al., 2005; Pesántez et al., 2018; Wilcke et al., 2001).

A seemingly increasing trend in the concentration of P from the headwaters to the main stem of the basin towards its outlet (Fig. 6c) could be attributed to the effect of anthropogenic activities in the study area. The cultivation of exotic species, particularly of cacao and passion fruit plantations, is common across the Cube catchment. Even though this practice is an important economic activity in the region, the low P concentrations in stream water, even at the middle and lower parts of the catchment (Fig. 6c), suggest a relatively low use of organic fertilizers. Nevertheless, future studies in the region should assess the concentrations of other substances used in cultivation, including pesticides, herbicides, and inorganic compounds to assess the impacts of this anthropogenic activity on stream water quality.

## 5.4 Mechanistic understanding of streamflow generation in intermittent and perennial tropical streams

**Figure 8** shows a conceptual diagram of the main water flow paths influencing streamflow generation in intermittent and perennial streams of the nested monitoring system of the Cube River catchment. From a process-based perspective, our findings indicate that shallow subsurface flow paths that mobilize water primarily through the thin litter layer (not shown in the diagram for simplicity) and the organic horizon of the soil under primary and secondary forests mainly located at the headwaters of the catchment dominate streamflow generation in intermittent streams **(Fig. 8a)**. Those shallow subsurface water flow paths are favoured by the quasi-impermeable nature of the underlying bedrock of the Playa Rica formation that reduces subsurface water storage and thus lead to a cessation of flow during the dry season (July-December). Differently, subcatchments draining perennial streams possess a high-water storage capacity that is replenished during the wet period (January to May) due to their moderate to high bedrock permeability **(Fig. 8b)**. Such recharge of groundwater helps sustain streamflow generation year-round in perennial streams despite the limited contribution from the litter layer and the organic horizon of the soil that has been substantially reduced or completely removed due to deforestation and cultivation in the Chocó-Darién of northwestern Ecuador. It is also noteworthy that there is little to no influence of surface or overland flow on streamflow generation in both types of streams, demonstrating the key role soil and geology play in streamflow dynamics and flow generation in this intermittent system of tropical streams.





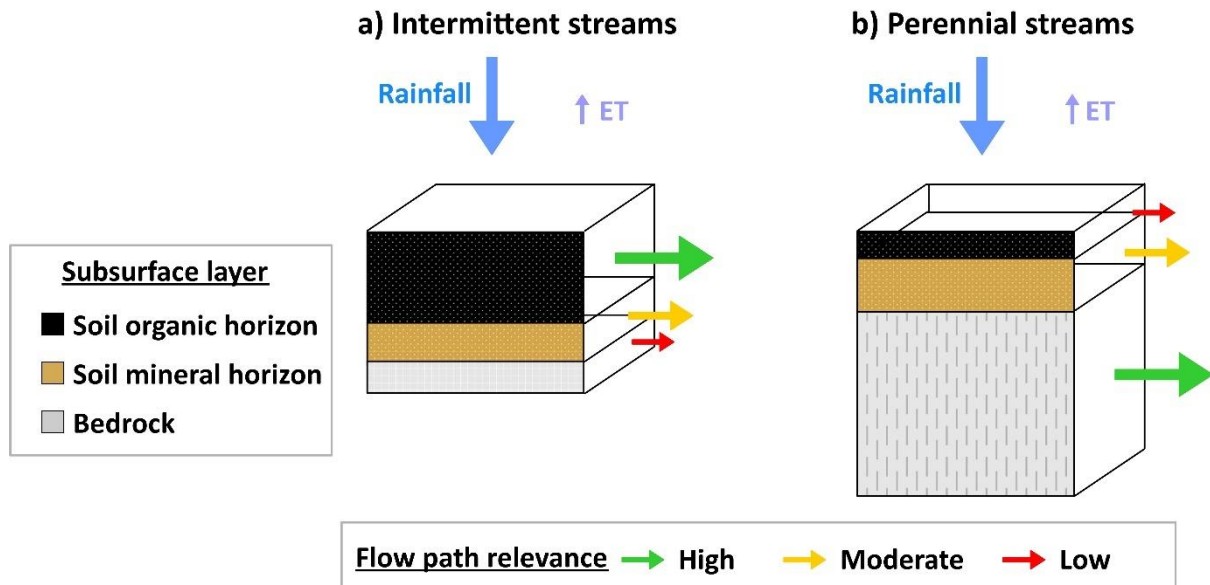

**Figure 8: Conceptual model of subsurface water flow paths influencing streamflow generation in a) intermittent and b) perennial tropical streams under changing land use at the Cube River catchment, northern Ecuador. The white surface at the top of the boxes represents the surface at the ground level. The size of the coloured boxes representing the soil and bedrock subsurface layers indicates their relative capacity to store water below ground. The dashed vertical lines in the bedrock layer of subplot b) denote the high bedrock permeability of the underlying geology at subcatchments draining perennial streams, as opposed to the almost impermeable bedrock underlying intermittent subcatchments depicted in a). Overland flow has not been observed in intermittent or perennial streams during field sampling campaigns, and thus, such a streamflow generation mechanism is neglected in the conceptual diagrams. The size and colour of the horizontal arrows represent the relative relevance of various water flow paths influencing streamflow generation.**

## 6 Conclusions

One of the key questions regarding the hydrology of intermittent hydrological systems is to determine how the interplay among hydrometeorological conditions, catchment surface and subsurface features, and land use influence streamflow dynamics and flow generation. Our nested monitoring approach comprising 20 spatially distributed streams (<1-159 km$^2$) within the Cube River catchment allowed to identify how landscape biophysical features influence streamflow dynamics in intermittent and perennial streams in a biodiversity hotspot on the Ecuadorian forest of the Chocó-Darién ecoregion. Our analysis indicates that the distinctive bedrock permeability of the geological formations found in the study area is the most important factor influencing flow generation. Our findings supported in a multimethod approach are key to highlighting the importance of carrying out detailed spatially distributed measurements across seasons to unravel the factors influencing the activation of different water flow paths and achieving a sound mechanistic understanding of flow processes in intermittent hydrological systems. While hydrometric data identifies differences in hydrological dynamics between intermittent and perennial streams, isotopic and geochemical data permits defining how water transports via surface and subsurface water flow paths. Complementarily, the relation between biophysical landscape characteristics and geochemical signals across the



nested system helps to define how intrinsic landscape characteristics influence the catchment's hydrological behaviour.
Locally, our findings provide key information for sustainable water management and climate change adaptation in the Chocó-Darién ecoregion. Beyond this, the study highlights the relevance of considering the landscape subsurface characteristics in the hydrological modelling of intermittent systems of streams. Taking advantage of the demonstrated potential of isotopic and geochemical tracers to investigate streamflow processes in this tropical intermittent hydrological system, further hydrological investigations could be targeted to determine the source and age of stream water in these
systems. In addition, future studies in the region should also assess how the identified streamflow dynamics and differences in water flow paths between intermittent and perennial streams influence biological and ecological processes in the ecoregion. Obtaining this information is crucial for improving the management and conservation of terrestrial and aquatic ecosystems and biodiversity in the region, as well as to secure a sustainable provision of ecosystem services to local communities that are particularly vulnerable to changes in regional land use and global climate patterns.

**Data availability**

The data used in this study are property of the Deanship of Research and Creativity at the Universidad San Francisco de Quito. These data will become publicly available in accordance with the rules and embargo regulations of the USFQ, but it is not yet known where the data will be hosted (please contact the corresponding authors for updates).

**Acknowledgments**

G.M.M. was supported by a Postdoctoral Fellowship from the Universidad San Francisco de Quito USFQ and the H2020 European Research and Innovation action Grant Agreement 869226 (DRYvER). We thank the Ecuadorian Ministry of Environment, Water, and Ecological Transition (MAATE) and particularly to the Reserva Ecological Mache-Chindul REMACH for providing research permits to conduct this study (MAAE-ARSFC-2020-1057). Thanks are due to Thibault Datry for his contribution to the study monitoring scheme, to Carla Villamarín, Segundo Chimbolema, Ricardo Jaramillo,
Diego Mosquera, and Karla Barragán for their fieldwork support, to Christian Suárez for contributing to the spatial analyses, and to Melany Ruiz-Urigüen and Natalia Carpintero for assessment in laboratory analyses. We also thank the logistical and field support of the staff of the Fundación para la Conservación de los Andes Tropicales (FCAT), particularly to Domingo Cabrera, Julio Loor, Darío Cantos, Luís Zambrano, Jorge Olivo, Carlos Aulestia, and Luis Carrasco.



**Financial support**

The research was funded by the European Union's Horizon 2020 research and innovation programme under grant agreement
No 869226 and the Universidad San Francisco de Quito USFQ through the project "Securing biodiversity, functional
integrity and ecosystem services in DRYing riVER networks project (DRYvER)".

**Author contribution**

Conceptualization and Methodology: G.M.M., A.E.; Formal analysis and Literature Review: G.M.M.; Investigation:
G.M.M.; Funding acquisition, Project administration, and Resources: A.E.,; Visualization: G.M.M., J.D.; Writing - original
draft: G.M.M.; Writing - review & editing: all authors. All authors have read and agreed to the published version of the
manuscript.

**Competing interests**

The authors declare that they have no conflict of interest.

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
