# Peer review of "Streamflow generation in a nested system of intermittent and perennial tropical streams under changing land use"

_EGUsphere, 2025_

## Author Response (AR1)

**Response to comments from the editor**

Dear authors,

Thanks for the detailed reply to the reviewer comments. As you can see, the reviewers are in general supportive of your manuscript, but made many suggestions and comments, which need to be addressed carefully in the revised version. I am also struggling a bit with the fact, that no meteorological data is available. Maybe in addition to the proposed climatological data from the monitoring station of the Ecuadorian National Agency of Hydrology and Climate (INAMHI), you should also try to get meteorological data from this station in order to link the observed responses to the meteorological focusing - this is very important.

Please provide a detailed response of the changes you did in the manuscript including a version with track changes.

Best regards

Markus Weiler

Reply: Dear Editor, we thank you for managing our manuscript. We have implemented all the reviewers' comments, except for those related to the local meteorological conditions. Unfortunately, as in many tropical regions around the planet, there was a complete lack of hydrometeorological data before we started working there. We were able to install water level probes during our study to characterize the streamflow dynamics, but due to limited funding and logistical constraints, we could not measure precipitation and other meteorological variables. To cope with this data limitation, we tried using remote sensing data, but no product was able to represent the conditions that we observed in the monitored streams. We suspect that the latter is likely related to the coarse spatial scale of such products, which cannot mimic the effect of the fine-scale rugged and variable topography affecting local climatology within the catchment as reported for tropical montane regions (Buytaert et al., 2006). At the same time, we could not access historical data from the INAMHI for the study year, as the update of this information has been delayed by the pandemics that affected the world in 2020-2021. For these reasons, it is impossible to characterize the climatology of the system for the study year. Nevertheless, the historical data from INAMHI allowed us to provide a sound characterization of climatological patterns in the region that cause hydrological intermittency. In addition, since our conclusions are based on different sources of data that support stream water hydrometric observations without depending on climatological conditions, we are convinced that we provide sufficient evidence to support our findings. In the following our responses are shown in **blue text** to the reviewers' comments presented in **black text**.

**Response to comments from Anonymous Referee #1**

Overview of Anonymous Referee #1:

This paper describes a unique dataset of water levels, chemistry, and isotopic composition of streamwater across a tropical catchment. The sampling design was such that it included intermittent and perennial streams. The results highlight the important role of geology and thereby subsurface storage capacity in determining which streams are intermittent and perennial. The differences in flow pathways in turn affect the stream chemistry.

The dataset is unique and the paper provides clear evidence for the role of geology in determining the variation in runoff responses and chemistry across the catchment, improving not only our knowledge of intermittent streams but of catchment functioning in general. The figures are all very clear and the paper is logically structured. At some places the writing can be a bit clearer or shorter (suggestions are given in the annotated pdf) but overall the text is very clear.

In my opinion the paper puts a bit too much emphasis (in its writing) on understanding intermittent streams as only some of the streams are intermittent and there is knowledge gained for the whole catchment. In other words, it is not wrong to focus so much on the intermittent streams but the paper actually provides more information than just for intermittent streams. This can however easily be solved with some rewording. Having said that, I would prefer to see an extra table regarding the statistical differences in the chemistry for the intermittent and perennial streams for the different sampling dates. Now only the correlations with land cover and geology are given.

Reply: We appreciate the reviewer's general perspective on the value of our work and thank him/her for the valuable feedback. In the revised manuscript we have addressed each of his/her observations/suggestions to improve quality of the manuscript. We also include a new figure to show the statistical differences of the chemical data (Figure 7 in the revised version of the manuscript). Please refer to our reply to this comment below for a detailed response. We have also implemented the suggested changes in the annotated pdf file to improve the readability of the manuscript.

Specific comments:

L98: I think that it would be useful to add a specific hypothesis here (or some research questions). If land use has the largest effect on streamflow generation and land degradation leads to more overland flow, one would expect that flow paths are shallower, leading to less recharge and faster responses and recessions and less perennial flow for

the lower parts of the catchment. If geology is the dominant factor, then soil depth and storage would determine which streams respond first and have faster recessions – which is indeed what the results show. The intro could highlight the effects of land degradation and soil depth/storage capacity on the runoff response a bit more by contrasting them and referring to some studies that have specifically looked at these factors.

Reply: Thank you for the suggestions. We have added two working hypotheses regarding the effects of land degradations and geology on streamflow generation in lines 106-109: "Our working hypotheses are as follows: i) if land use has the largest effect on streamflow generation, overland flow would increase in highly deforested catchments thus leading to diminished subsurface recharge and less perennial flow; whereas ii) if geology is the dominant factor, the catchments with high subsurface water storage would generate flow perennially and intermittent flow would occur in catchments with low subsurface water storage.".

In the revised version of the manuscript, we highlight the effect of land degradation and soil depth/storage capacity on the runoff response in the introduction in lines 68-73: "Through hydrological modelling, Gutierrez Jurado et al. (2021) pinpointed that soil type and hydraulic properties can exert great control on the mechanisms driving flow generation in a Mediterranean coastal catchment in South Australia by influencing unsaturated subsurface water storage. Regarding changes in land use, in a strongly seasonal Mediterranean region in the Central Spanish Pyrenees deforestation caused a change in streamflow hydrological dynamics due the loss of forest cover, including higher peak flows and higher low flows, compared to a conserved forested catchment (Serrano-Muela et al., 2008).".

Section 2: Some important information about the catchment is missing, such as the annual precipitation, temperature (and potential ET). Also, it would be good to already mention here the differences in soil depth and permeability across the catchment

Reply: No climatological information was available during the study period in the study area, and we were not able to install precipitation gauges due to limited funding and logistical constraints. Historical precipitation records of a gauging station of the Ecuadorian National Agency of Hydrology and Climate (INAMHI) located approximately 20 km from the study site are now presented to characterize the precipitation regime in lines 125-126: "Mean annual precipitation at a gauging station of the Ecuadorian National Agency of Hydrology and Meteorology (INAMHI) located approximately 20 km from the study area at 100 m a.s.l. during the period 2005-2017 is 1256 mm yr-1.".

Although no quantitative information about soil depth and permeability across the whole catchment is available due to its remoteness and limited access because of the steep topography, the characteristics of the two geological formations are well differentiated qualitatively through regional geological prospections. We now provide further details about the geological features of the catchment in lines 152-156: "The Playa Rica formation covers 11% of the catchment and is primarily found in headwaters areas (Fig. 1d). This formation dates from the Oligocene period (23-33 Ma). It possesses a very low bedrock permeability with a thickness of 800 m (DIIEA, 2010) and its lithology is composed of shales and sandstones. The younger Viche formation dating from the Miocene (5-23 Ma) has a medium bedrock permeability with a thickness of approximately 400 to 1000 m."

Section 3.4: The analysis method for COD needs to be described here as well.

Reply: Information about the COD analysis method has been added in lines 241-242: "COD was analysed using the 20-1500 mg/L COD (HR) Hach kit following the Standard Method (SM) 5220 D (APHA, 2023).".

L277: This sounds like the perennial streams also went dry. Is that correct? Then they are not perennial. This is a bit confusing.

Reply: Perennial streams did not dry. The sentence was rephrased as follows in lines 292-294: "At a yearly time scale, the hydrographs show that intermittent streams (Fig. 2a) returned to zero-flow faster after rainfall cessation than perennial streams returned to baseflow (Fig. 2b). The hydrographs also show that perennial streams had longer recessions than the intermittent streams.".

L283: Provide some information on the size and intensity of the selected events. Are they similar for the wet and dry period example?

Reply: Unfortunately, this is not possible as there were no rainfall data available during the study period. We report the dynamic of streamflow response during the wet and dry periods when such response was simultaneously detected at both intermittent and perennial streams in lines 203-205: "To identify potential differences in hydrological dynamic during the wet and dry periods, we analysed the hydrological response during events that presented a simultaneous water level response for both stream types during each period."

Section 4.1: I would make it much clearer that the results from WETSPRO are based on graphical hydrograph analyses. That way it clearly distinguishes them from the isotope or tracer based inferences of flow pathways.

Reply: We agree. This is now mentioned in lines 294-295: "The graphical separation of the hydrographs into their subcomponents quantitatively depicts clear differences in the hydrological dynamic of both types of flow regimes."

L319-321: Is this difference statistically significant?

Reply: Yes, the difference is statistically significant. Differences between the datasets were tested using the Wilcoxon rank- sum test (Wilcoxon, 1945) and this is now specified in the methods section in lines 226-228: "The Wilcoxon rank-sum test (Wilcoxon, 1945) was used to assess differences between the stream water isotopic composition of intermittent and perennial streams at a 95% confidence level (i.e. p-value < 0.05) given that data were non-parametric.". The p-value < 0.05 of the test suggests a significant difference between both datasets, and this is now reported in line 359-331: "During the wettest period, the isotopic composition of $\delta 18O$ in the smallest subcatchments (S1-S8) was significantly (p-value < 0.05) depleted (mean±standard deviation: -4.7±0.3‰) compared to sampling sites with larger drainage areas (-4.1±0.3‰) (Fig. 4a).".

L379: I fully understand the need to focus on a few solutes but it is not clear to me why you used P as an indicator of agriculture. Are there high additions of P in these agricultural systems? Provide a bit more background information on this. Why not base the selection of the solutes on the test for which solutes the differences between catchments with different land uses are statistically significant or the correlation between % agriculture and concentrations (i.e., Table 3)?

Reply: Fertilizers based on phosphate rock help control the acidity in soils with low pH. Given that the soils in the study region tend to be acidic, phosphate rock based fertilizers are preferred for local cultivation practices. This is the reason why we choose P as an indicator of anthropogenic activities, particularly agricultural practices, in the study region. This is now specified in lines 386-389: "Since the addition of low-cost phosphate rock-based fertilizers is preferred in local cultivation practices due to the acidic nature of the local soil, the concentration of P was used as an indicator of how changes in land use due to anthropogenic activities could potentially affect the hydrological behaviour and chemical characteristics of water at the study site."

We did not base the selection of the solutes on the statistical significance between the solutes' concentrations and the catchments' features as no statistically significant correlations were found for most catchments' features, including agriculture, as shown in

Table 3. The statistical significance of the correlation analysis was rather used to assess whether there is a relation between the solutes concentrations and the factors influencing the catchments' hydrological behavior.

Figure 6: Perhaps it is useful to use two different colors to indicate which sites are intermittent and perennial. Overall, I would like to see a bit more information on the (statistical) differences in chemistry between the intermittent and perennial streams as this is a unique part of the dataset. Perhaps add information on the significance of the differences in concentrations for intermittent and perennial streams in Table 3 or add another table where you show this for each sampling date.

Reply: We now use two different colors in Figure 6 as shown below to indicate the sites that are intermittent (in light yellow) and perennial (in light blue).

[Figure]

In the revised version of the manuscript, we include a new figure (Figure 7) presented below to show the differences in the water chemistry between intermittent and perennial streams and use the Wilcoxon rank-sum test (Wilcoxon, 1945) to assess the statistical significance of the differences. This has been specified in the methods section in lines 268-269: "Differences in stream water chemistry were assessed using the Wilcoxon rank-sum test (Wilcoxon, 1945) at a 95% confidence level (i.e. p-value < 0.05) since data were non-parametric.". The results of this analysis are shown in lines 398-400: "The Wilcoxon rank sum test indicates that the concentrations of Ca (Fig. 7a) and P (Fig. 7c) are significantly different (p-value < 0.05) between intermittent and perennial streams, while no statistically significant difference can be assumed for TOC (p-value = 0.18; Fig. 7b)."

[Figure]

**Figure 7: Boxplots of the concentrations of Calcium (Ca), total organic carbon (TOC), and phosphorus (P) for the intermittent (Int in light yellow) and perennial (Per in light blue) streams monitored within the Cube River catchment during six monitoring campaigns carried out in 2021. The horizontal lines represent the first, second (median) and third quartiles, the whiskers represent 0.4 times the interquartile range, and the black dots are outliers.**

Figure 7: I don't think that you need the 3D surface plot, you can just color-code the points in b and c according to Viche formation and elevation, respectively. If you want to keep the 3D surface plot, then at least show the points as well. Also, why not use a multiple linear model to look at the combined effect of the geology and elevation (as they are somewhat correlated). And why not show the same results/graphs for the other two chosen solutes a well?

Reply: We agree the surface plot was not needed, so it was removed.

[Figure]

Figure 1: Correlation between a) mean altitude and Ca concentration and b) between the areal extent of the Viche formation and Ca concentration. Note that the color scheme in a) is only used to visualize the interplay between the predictor variables (mean altitude and Viche formation) and the response variable Ca concentrations.

Also, it is not necessary to color-code the plots, as the correlations between Ca and both the Viche formation and vegetation are shown. We did not use a multiple linear model as we found a high statistically significant correlation between geology and elevation (section 4.4) that could cause a model overfitting issue (Lin et al., 2011). Also, we do not show the scatter plot of the other chosen solutes as correlation between them, and the features of the landscape are low and non-statistically significant as shown in Table 3.

Section 4.4. The effects of land use and soil depth on the concentrations are not so well described and discussed. Also, it would be very useful to mention the correlation between geology and mean catchment elevation.

Reply: We now specify that "Land cover, soil type, and mean slope showed weak and non-statistically significant correlations with all solutes (Table 3)." In lines 425-426. The correlation between geology and mean catchment elevation was already reported in the original draft as follows: "... given a strong, negative, and significant correlation (r=-0.79) between mean altitude and the Viche formation (not shown for brevity).", see lines 419-420 in the revised manuscript.

L434-443: This is a bit of a selective comparison. I don't think that it was the ET itself that was the cause for the differences. This study (figure 2) and the study by Zimmer and McGlynn both show that antecedent conditions are important. In the study by Zimmer and McGlynn the differences in antecedent conditions are due to seasonal changes in ET, whereas in your catchments they are likely mainly due to seasonal differences in precipitation. Also, the fact that there is no evaporative fractionation for the water in the stream does not mean that there is no effect of ET on the antecedent conditions and the relative importance of different runoff processes. Therefore, this section requires some rewording. Also, L441-453 are interesting but this is certainly not the main and most

important outcome of the study. I am not sure that you need this text and think that it distracts from the main results (especially when it is the first thing in the discussion).

Reply: We thank the reviewer for pointing this out. We agree that antecedent conditions are important in our study as in the referred paper (Zimmer and McGlynn, 2017). This is now specified in lines 444-459: "High flows with a flashy response during the wet season and low flows with longer recessions during the dry season (Fig. 2a-2d) suggest that antecedent wetness conditions are key to drive the hydrological dynamics of streamflow at our study site. These dynamics are supported by the depletion of δ18O in stream water during the wet season (Fig. 3), likely related to the input of high rainfall amounts depleted in the heavy stable isotopes of water (e.g. Mosquera et al., 2016a). Similar hydrological dynamics were observed at an intermittent low relief and highly weathered catchment with a subtropical climate in North Carolina, USA (Zimmer and McGlynn, 2017). However, while in the USA catchment antecedent wetness was controlled by a seasonal change in evapotranspiration with little temporal variability of precipitation; it is likely that the strong seasonal variation of precipitation is the main factor driving the temporal changes in antecedent wetness within the Cube River catchment. The latter likely results from a combined effect of the high rainfall during the wet period and the continuous cloudy and foggy conditions during the dry season. These hydrometeorological conditions likely reduce water losses to the atmosphere via evapotranspiration as has been observed in other tropical montane catchments in Ecuador (e.g. Mosquera et al., 2024; Ochoa-Sánchez et al., 2020). In turn, these conditions help to maintain the streambed humid throughout the year, even in intermittent streams. In addition, the virtually negligible spatial and temporal variation of d-excess across our nested monitoring system throughout the year (Figs 4c, 4d) further emphasizes the smaller role evapotranspiration has on the streamflow dynamics of intermittent and perennial streams as compared to precipitation seasonality.".

Also, we agree that L441-453 were not relevant for the study and the paragraph was removed from the discussion as suggested.

L465-470: You may want to highlight that in addition to contact time, there could also be a difference in the reactivity of the material of the two formations.

Reply: This is possible and has been added in lines 479-481: "... (Figs. 5,6a) support this finding and indicate a higher contact time of water with the underlying bedrock and/or higher reactivity of the geologic material."

L533: Mention this already in section 5.3.

Reply: We mention this in lines 514-516 in section 5.3: "It is also worth noting that regardless of the degree of deforestation, we did not observe an extensive occurrence of overland flow throughout the Cube River catchment during the monitoring campaigns."

**Response to comments from Anonymous Referee #2**

Overview of Anonymous Referee #2:

This is a well written and timely manuscript looking at the various controls of streamflow generation in a tropical catchment. The work on the intermittent streams is particularly important as this is an area of need, especially in the tropics where work is limited. The authors have put in good work and I only have some minor suggestions.

Reply: We appreciate the reviewer's general impression on the value of our manuscript. We have updated the manuscript considering his/her valuable feedback. Below we addressed all the reviewer's comments and suggestions.

L113-114: It will be good to include the long term average rainfall totals for each period.

Reply: We now include historical data from one monitoring station of the Ecuadorian National Agency of Hydrology and Climate (INAMHI) located at approximately 20 km from the study site to provide a general description of the climatology of the region, indicating the mean annual rainfall and its distribution during the wet and dry seasons in lines 125-128: "Mean annual precipitation at a gauging station of the Ecuadorian National Agency of Hydrology and Meteorology (INAMHI) located approximately 20 km from the study area at 100 m a.s.l. during the period 2005-2017 is 1256 mm yr-1. Seventy-five percent of precipitation is concentrated in the wet period extending from January to May and 25% is distributed during the dry period from June to December.        ".

L125: Please include a reference for the soil types.

Reply: A reference for the soil types has been included in lines 136-137: "According to the USDA Soil Taxonomy (Soil Survey Staff, 2022), the soils are classified as inceptisols, malisons, entisols, and miscellaneous".

L284-285: Please include a table showing the response characteristics. This can be included as a supplementary file.

Reply: The information was presented in the boxes of figure 2 in the manuscript of the original submission. We consider it is useful to keep it there and this is now specified in the figure caption in lines 311: "Results from the graphical hydrograph analysis are written in the boxes of each subplot.".

L300: This section is well written. The authors do a good job of showing the isotopic differences based on seasonal changes. However, I think that it will benefit the reader if we could see this isotopic evolution for a perennial vs intermittent stream. It will be good to see if there are major differences particularly in the transition periods.

Reply: The figure now shows the isotope ratios of the intermittent and perennial sites using different symbols.

[Figure]

**Figure 2: Relationship between oxygen-18 ($\delta^{18}O$) and hydrogen-2 ($\delta^2 H$) isotopic ratios in stream water samples collected at intermittent (circles) and perennial (triangles) sites within the Cube River catchment during six monitoring campaigns (M1-M6) carried out during the wet (M1 and M2), transition (M3) and dry (M4, M5, and M6) periods in 2021. Monitoring campaigns M2 and M6 correspond to the wettest (dark blue symbols) and driest (red symbols) sampling periods, respectively.**

This sentence has been added to the manuscript in lines 320-322 in this regard: "From the beginning of the wet season (M1) to the early dry season (M4) the isotopic ratios of the intermittent streams tended to be smaller compared to the perennial streams (Fig. 3); while the isotopic composition in both intermittent and perennial streams varied in a similar range in the middle and late dry season (M5 and M6)."

Figure 6: Can you indicate the perennial vs intermittent streams on this? I think it will also be good to remind readers that the numbers go from headwaters to the main outlet. Why was the Ca not plot as ppm like in all other figures.

Reply: We now show the perennial and intermittent sites in light blue and light yellow colors as shown above. The units in the figure were updated to ppm. We also indicate the readers what the order of the numbers is based on in lines 406-409: "In the x-axis, the sampling sites are ordered according to the elevation of their outlets, generally corresponding to the sites with the smallest drainage areas to be located at the Cube River headwaters and the sites with the largest drainage areas downstream toward the catchment's outlet (Fig. 1a, Table 1)."

[Figure]

Table 3: blue are positive correlations and red are negative. Please indicate this so that it is easier to recognize from the table.

Reply: Colors in the tables are not allowed as per the journal guidelines, so we do not use the blue/red classification in the revised version of the manuscript to differentiate between positive and negative correlations. High and statistically significant correlations are now underlined and in bold in Table 3.

| Element | Drainage area (km²) | Mean altitude (m a.s.l.) | Mean slope (%) | Land cover[a] (%) | | | | | Distribution of soil types[b] (%) | | | | Geology[c] (%) | |
|---|---|---|---|---|---|---|---|---|---|---|---|---|---|---|
| | | | | NF | MF | AM | WB | BG | MOL | ENT | INC | MIS | PR | VI |
| Ba | 0.18 | -0.58 | 0.25 | -0.14 | 0.07 | 0.27 | -0.06 | -0.32 | -0.46 | 0.20 | 0.43 | -0.28 | -0.44 | 0.44 |
| Ca | 0.69 | **-0.87** | 0.48 | -0.53 | 0.38 | 0.54 | 0.47 | -0.41 | -0.28 | 0.64 | 0.23 | 0.29 | **-0.78** | **0.78** |
| K | 0.63 | **-0.83** | 0.50 | -0.47 | 0.44 | 0.49 | 0.45 | -0.37 | -0.41 | 0.53 | 0.37 | 0.19 | **-0.75** | **0.75** |
| Mg | 0.62 | **-0.80** | 0.50 | -0.38 | 0.32 | 0.42 | 0.38 | -0.36 | -0.36 | 0.56 | 0.31 | 0.22 | -0.74 | 0.74 |
| Mn | -0.17 | -0.19 | -0.15 | -0.06 | 0.22 | -0.06 | 0.10 | 0.09 | -0.28 | 0.31 | 0.08 | -0.02 | -0.38 | 0.38 |
| Na | 0.67 | **-0.78** | 0.53 | -0.40 | 0.30 | 0.47 | 0.41 | -0.35 | -0.30 | 0.43 | 0.27 | 0.13 | -0.64 | 0.64 |
| Pb | 0.44 | -0.50 | 0.33 | -0.54 | -0.08 | 0.53 | 0.42 | -0.43 | -0.16 | 0.47 | 0.11 | 0.33 | -0.56 | 0.56 |
| Alk | 0.63 | **-0.83** | 0.52 | -0.52 | 0.32 | 0.53 | 0.47 | -0.43 | -0.38 | 0.52 | 0.35 | 0.24 | **-0.76** | **0.76** |
| F | 0.69 | **-0.85** | 0.51 | -0.41 | 0.43 | 0.42 | 0.46 | -0.35 | -0.29 | 0.66 | 0.20 | 0.29 | **-0.79** | **0.79** |
| SO4 | 0.55 | **-0.85** | 0.31 | -0.33 | 0.35 | 0.43 | 0.24 | -0.38 | -0.32 | 0.53 | 0.30 | 0.08 | -0.64 | 0.64 |
| COD | 0.48 | **-0.76** | 0.19 | -0.51 | 0.37 | 0.42 | 0.52 | -0.12 | -0.24 | **0.76** | 0.08 | 0.26 | -0.74 | 0.74 |
| P | 0.65 | **-0.81** | 0.41 | -0.55 | 0.32 | 0.44 | 0.53 | -0.22 | -0.16 | 0.72 | 0.02 | 0.39 | **-0.86** | **0.86** |
| TOC | 0.04 | 0.17 | -0.19 | 0.04 | 0.23 | -0.13 | 0.10 | 0.30 | 0.13 | 0.05 | -0.08 | 0.25 | 0.14 | -0.14 |
| T | **0.76** | **-0.87** | 0.48 | -0.53 | 0.43 | 0.44 | 0.67 | -0.22 | -0.26 | 0.72 | 0.19 | 0.49 | **-0.77** | **0.77** |
| pH | **0.84** | -0.70 | 0.51 | -0.43 | 0.24 | 0.47 | 0.49 | -0.20 | 0.06 | 0.52 | 0.01 | 0.37 | -0.48 | 0.48 |
| EC | 0.66 | **-0.86** | 0.48 | -0.44 | 0.41 | 0.47 | 0.41 | -0.38 | -0.34 | 0.54 | 0.30 | 0.20 | **-0.77** | **0.77** |
| DO | **0.85** | -0.56 | 0.54 | -0.32 | 0.25 | 0.34 | 0.49 | -0.12 | 0.14 | 0.45 | -0.03 | 0.49 | -0.34 | 0.34 |

L454-455: In this section is it a bit unclear how the subsurface features affect the streamflow dynamics given that some of the gauges were located in different altitude and land use covers. This is especially true for the Viche formation which only had one intermittent stream which makes it difficult to compare to the other intermittent streams of the other geological formation.

Our analysis is based on a nested monitoring system approach given that our objective is to identify general patterns of the hydrological response of intermittent and perennial catchments, rather than using a comparison of paired catchments with very specific conditions that could obscure such patterns. Therefore, we consider that including catchments with variable size, mean elevation, and landscape features is an advantage of our approach, rather than a weakness. Here, we would also like to highlight that the use of the presented multimethod approach has the advantage of avoiding subjectivity when inferring the hydrological behavior of the catchments as opposed to relying on a single source of information (i.e., stable isotopes vs. geochemical data vs. hydrometric data) that could lead to high uncertainties in the identification of the mechanistic understanding of streamflow generation. Because of this, we consider that our study should remain focused on the presented spatially distributed measurements across spatial scales and landscape characteristics, rather than assessing specific differences among the catchments.

L491: Were there any perennial streams in forested areas? If so what was the geochemistry like when compared to the intermittent streams?

Reply: Yes, site S2 corresponds to a forested catchment with perennial flow. It had high TOC, like other sites dominated by forest cover; relatively high P as other perennial sites; but relatively low calcium compared to other perennial sites. This is now mentioned in lines 357-358: "Even though the largest concentrations of P and TOC were observed at a forest headwater catchment generating perennial flow (S2)".

Overall I would like to see a bit more of a comparison of the one intermittent stream located in the Viche geology when compared to the other locations.

Reply: Unfortunately, the intermittent site located on the Viche geology had no flow during the three sampling campaigns carried out during the dry season as reported in the caption of table 2 in lines 368-369 "... site S14 (marked with † symbol) completely dried during three sampling campaigns in the dry season". Therefore, it is not possible to directly compare it to the other locations. However, we would like to emphasize once more that our aim is not to focus on specific catchment features but take advantage of the multimethod dataset applied to the nested system of catchments to understand general patterns of flow generation in intermittent versus perennial streams.

**Response to comments arising from the manuscript validation process**

Some your tables contain coloured cells or/and coloured values. Please note that this will not be possible in the final revised version of the paper due to HTML conversion of the paper. When revising the final version, you can use footnotes or italic/bold font.

Reply: In the revised version of the manuscript, we no longer include colored cells or values in the tables. Values we want to highlight are now displayed in bold, italics, or underlined.

**REFERENCES WE CITED**

APHA: Standard methods for the examination of water and wastewater, 24th ed., edited by: Lipps, W. C., Braun-Howland, E. B., and Baxter, T. E., American Public Health Association (APHA), American Water Works Association (AWWA) and Water Environment Federation (WEF)). , 2023.

Buytaert, W., Celleri, R., Willems, P., Bièvre, B. De, and Wyseure, G.: Spatial and temporal rainfall variability in mountainous areas: A case study from the south Ecuadorian Andes, J Hydrol (Amst), 329, 413–421, https://doi.org/10.1016/j.jhydrol.2006.02.031, 2006.

Gutierrez-Jurado, K. Y., Partington, D., and Shanafield, M.: Taking theory to the field: Streamflow generation mechanisms in an intermittent Mediterranean catchment, Hydrol Earth Syst Sci, 25, 4299–4317, https://doi.org/10.5194/HESS-25-4299-2021, 2021.

Lin, D., Foster, D. P., and Ungar, L. H.: VIF Regression: A Fast Regression Algorithm for Large Data on JSTOR, J. Am. Stat. Assoc., 106, 232–247, https://doi.org/https://doi.org/10.1198/jasa.2011.tm10113, 2011.

Mosquera, G. M., Célleri, R., Lazo, P. X., Vaché, K. B., Perakis, S. S., and Crespo, P.: Combined Use of Isotopic and Hydrometric Data to Conceptualize Ecohydrological Processes in a High-Elevation Tropical Ecosystem, Hydrol Process, https://doi.org/10.1002/hyp.10927, 2016a.

Mosquera, G. M., Marín, F., Carabajo-Hidalgo, A., Asbjornsen, H., Célleri, R., and Crespo, P.: Ecohydrological assessment of the water balance of the world's highest elevation tropical forest (Polylepis), Science of The Total Environment, 941, 173671, https://doi.org/10.1016/J.SCITOTENV.2024.173671, 2024.

Ochoa-Sánchez, A. E., Crespo, P., Carrillo-Rojas, G., Marín, F., and Célleri, R.: Unravelling evapotranspiration controls and components in tropical Andean tussock grasslands, Hydrol Process, 34, 2117–2127, https://doi.org/10.1002/hyp.13716, 2020.

Serrano-Muela, M. P., Lana-Renault, N., Nadal-Romero, E., Regüés, D., Latron, J., Martí-Bono, C., and García-Ruíz, J.: Forests and Their Hydrological Effects in Mediterranean Mountains, 28, 279–285, https://doi.org/10.1659/MRD.0876, 2008.

Soil Survey Staff: Keys to Soil Taxonomy, 13th ed., USDA Natural Resources Conservation Service., Washington, DC, 1–401 pp., 2022.

Wilcoxon, F.: Individual Comparisons by Ranking Methods, Biometrics Bulletin, 1, 80, https://doi.org/10.2307/3001968, 1945.

Zimmer, M. A. and McGlynn, B. L.: Ephemeral and intermittent runoff generation processes in a low relief, highly weathered catchment, Water Resour Res, 53, 7055–7077, https://doi.org/10.1002/2016WR019742, 2017.